# Surface paleothermometry using low temperature thermoluminescence of feldspar

Rabiul H. Biswas[1], Frédéric Herman[1], Georgina E. King[1], Benjamin Lehmann[1], Ashok K. Singhvi[2]

[1]Institute of Earth Surface Dynamics, University of Lausanne, Lausanne, Switzerland

[2]Atomic, Molecular and Optical Physics, Physical Research Laboratory, Ahmedabad, India

*Correspondence to*: Rabiul H. Biswas (biswasrabiul@gmail.com)

**Abstract**. Thermoluminescence (TL) of feldspar is investigated for its potential to extract temperature histories experienced by rocks exposed at Earth's surface. TL signals from feldspar observed in the laboratory arise from the release of trapped electrons from a continuous distribution of trapping energies that have range of thermal stabilities. The distribution of trapping

energies, or thermal stabilities, is such that the lifetime of trapped electrons at room temperature ranges from less than a year to several billion years. Shorter lifetimes are associated with low temperature TL signals, or peaks, and longer lifetimes are associated with high temperature TL signals. Here we show that trapping energies associated with shorter lifetimes, or lower temperature TL signals (i.e., between 200 °C and 250 °C), are sensitive to temperature fluctuations occurring at Earth's surface over geological timescales. Furthermore, we show that it is possible to reconstruct past surface temperature histories in

terrestrial settings by exploiting the continuous distribution of trapping energies. The potential of this method is first tested through theoretical experiments, in which a periodic temperature history is applied to a kinetic model that encapsulates the kinetic characteristics of TL-thermometry. We then use a Bayesian approach to invert TL measurements into temperature histories of rocks, assuming that past temperature variations follow climate variations observed in the $\delta^{18}O$ records. Finally, we test the approach on two samples collected at the Mer de Glace (Mont Blanc massif, European Alps) and find similar

temperature histories for both samples. Our results show that TL of feldspar may be used as a paleo-thermometer.

## 1 Introduction

Earth's climate fluctuates in a cyclic way, from years to million-year time scales driven by Earth's orbital processes and rare aberrant shift and extreme climate transients during the last $10^3$ to $10^5$ years (e.g., Zachos et al., 2001). Reconstructions of past terrestrial climates often rely on the use of climate proxies that preserve the physical and/or chemical characteristics related to

past climate of Earth's history. Examples of such proxies include ice cores, tree rings, sub-fossil pollen, boreholes, corals, lake sediments and carbonate speleothems (e.g., Jones and Mann, 2004 for a review). Although they have provided invaluable insights into past climates and their physical characteristics, very few of these proxies provide a direct measure of temperature variations in continental settings (e.g., glycerol dialkyl glycerol tetraether, GDGT; Tierney et al., 2012), and many of these methods often suffer from methodological limitations that limit reliable construction of terrestrial temperatures. For example,

fossil pollen and plant macrofossils have provided key insights into past terrestrial climate at millennial timescales (e.g., Bartlein et al., 2011), but typically rely on numerous additional climate parameters, including precipitation, plant-available moisture, seasonality, and the length of the growing season that make the inference of rock temperature histories challenging. Similarly, GDGTs rely on the preservation of organic compounds. To address some of these issues, Tremblay et al. (2014a, b) recently introduced a new paleotemperature proxy based on the thermal stability and release of $^3He$ and $^{21}Ne$ noble gases in

quartz. The system has a single energy level and is therefore only able to estimate a single equivalent diffusion temperature (EDT). As a result, reconstruction of complex rock temperature histories challenging.

This study revives the use of thermoluminescence (TL) as a paleothermometer and takes advantage of recent progress made for TL thermochronometry (Biswas et al., 2018) to introduce a new approach for reconstructing temperature histories in

terrestrial settings at millennial timescales. The idea is not new; it has already been tested through a series of feasibility studies (Li and Li, 2012; Ronca, 1964; Ronca and Zeller, 1965) and was applied to lunar samples (Durrani et al., 1977). Unfortunately, the method has remained underdeveloped since this early work. More recently, Guralnik and Sohbati (2019) used Optically Stimulated Luminescence (OSL) to estimate paleotemperature. However, they only investigated a single thermal stability, and thus a single equivalent temperature, similar to the noble gas approach of Tremblay et al. (2014a, b).

The present study aims to establish TL of feldspar as a paleothermometer to constrain temporal variations of Earth's surface temperature, within the last few tens of thousands of years. TL signals from feldspar arise from a series of traps that have different thermal stabilities; with lifetimes at room temperature ranging from <yr to >Gyr (Biswas et al., 2018). In the following, we first investigate the sensitivity of the TL signals to linear and periodic thermal histories. Then, we use a Bayesian approach to infer the thermal histories of rocks from TL measurements. Finally, we apply this method to two samples collected from the Mer de Glace glacier (Mont Blanc massif, European Alps) as proof-of-concept.

## 2 Theoretical background

In its simplest form, luminescence enables the measurement of the concentration of charge trapped in the impurity or defect centers of natural crystalline minerals (e.g., quartz or feldspar); the trapped charge population is proportional to the time elapsed since the material was either heated or exposed to sunlight. In the laboratory when the minerals are heated using a linear heating rate, trapped electrons are released from increasingly higher energy traps, some of which recombine radiatively with the holes, and the luminescence process generates a TL glow curve. Trapping of electrons happens in nature when samples are exposed to environmental radiation over geological time. Detrapping reflects the escape of electrons from traps, which can occur thermally or athermally. In nature, the TL system can reach a dynamic equilibrium between radiation induced growth and decay via thermal and athermal pathways. For higher ambient temperatures, the thermal decay rate is greater, which results in a lower equilibrium level and vice versa (for more details see Aitken, 1985 and Ronca, 1964). Such a difference in equilibrium can be exploited to reconstruct the thermal history of rocks (Biswas et al., 2018; Brown et al., 2017; Guralnik and Sohbati, 2019; Herman and King, 2018; Herman et al., 2010; King et al., 2016a; King et al., 2016b). Until now, this principle has only been exploited for inferring rock cooling due to exhumation. Here we use the same principle for inferring the paleotemperature of rocks exposed at Earth's surface. In this section, we start by outlining the theoretical model that describes the TL process of feldspar, and how can one constrain all the model parameters. Then we assess how sensitive the model is to various surface temperature histories and ultimately show how surface temperatures in terrestrial settings can influence TL signals.

### 2.1 TL kinetic model and TL thermometer

The kinetic model that describes TL of feldspar was reviewed in Biswas et al. (2018). The model encapsulates the process of populating traps with electrons in response to surrounding radiation, and the processes of electron escape through thermal and athermal pathways. A key element of the method is that each kinetic parameter can be constrained in the laboratory for each sample. We briefly outline the kinetic model here and illustrate how the kinetic parameters may be constrained from laboratory measurements for one sample. The reader is referred to the supplementary material S1 and Biswas et al. (2018) for further information.

The model assumes general order kinetics (Biswas et al., 2018; Guralnik et al., 2015a; Guralnik et al., 2015b). The rate equation for the trapped charge population of a specific trapping center, with single trap depth ($E$), frequency factor ($s$), and athermal fading parameter ($\rho'$; Huntley, 2006; Tachiya and Mozumder, 1974) is as follows:

$$\frac{d}{dt}\big(\bar{n}(r',t)\big) = \frac{\dot{D}}{D_0}\big(1 - \bar{n}(r',t)\big)^a - se^{-\frac{E}{kT}}\big(\bar{n}(r',t)\big)^b - \tilde{s}e^{-\rho'^{-\frac{1}{3}}r'}\bar{n}(r',t) \qquad (1)$$

where $\bar{n}$ is equal to $n/N$ (where $n$ is the number of trapped electrons at time $t$ and temperature $T$, and $N$ is the total number of available traps), $\dot{D}$ is the dose rate due to ambient radioactivity (Gy ka$^{-1}$), $D_0$ is the onset of dose saturation (Gy), $a$ and $b$ are the kinetic orders of trapping and thermal detrapping respectively, $E$ is the trap depth or activation energy (eV), $s$ and $\tilde{s}$ are the thermal and athermal frequency factor respectively (s$^{-1}$), $\rho'$ is the dimensionless athermal fading rate and $r'$ is a dimensionless distance that characterizes the probability of athermal escape (Huntley, 2006). Each of these parameters can then be constrained from laboratory experiments that are described in Biswas et al. (2018) and summarized in the supplementary material S1 for one sample.

To account for athermal loss, i.e., anomalous fading (Wintle, 1973), the total number of trapped electrons at any instant $\bar{n}(t)$ is obtained by integrating over the whole range of dimensionless distances ($0 < r' < 2$; Kars et al., 2008) over which electrons can athermally escape;

$$\bar{n}(t) = \int_0^\infty p(r')\bar{n}(r',t)dr' \qquad (2)$$

where $p(r')$ is the probability of nearest recombination center at a distance between $r'$ and $r' + dr'$ and expressed as $p(r')dr' = 3r'^2 e^{r'^3}dr'$ (Huntley, 2006). This model was validated using rocks from the KTB borehole and applied to samples from Namche Barwa (Biswas et al., 2018), which gave results in agreement with other studies from the same area (King et al., 2016a).

TL of feldspar arises from a continuous distribution of trapping energies (Biswas et al., 2018; Duller, 1997; Grün and Packman, 1994; Pagonis et al., 2014; Strickertsson, 1985) and can be assumed as sum of a large number of traps (Biswas et al., 2018; Pagonis et al., 2014); all follow the process described by Eq. 1. To constrain the kinetic parameters in Eq. 1, we measure full TL glow curves and see how the kinetic parameters are distributed along the TL glow curve (Biswas et al., 2018). For the modeling, we use the kinetic parameters for sample MBTP9 (Lehmann et al., 2020 for sample details). The experimental details are provided in section 4 and supplementary material S1. The distribution of kinetic parameters along the TL glow curve temperature is reported in Fig. 1. The results show that the kinetic parameters vary systemically with the glow curve temperature. Such data are then fitted using a spline function from which the kinetic parameters are then extracted for a specific TL temperature (Biswas et al., 2018), which we define herein as a specific "TL thermometer".

*Figure 1*

## 2.2 Temperature response of TL thermometers

In this section, we use the model (Eq. 1) in a forward manner by prescribing a temperature history and predicting the trapped charge population through time ($\bar{n}$). Both linear (isothermal, warming and cooling) and periodic thermal histories are used. The model was run for 1 Myr, with an initial condition of $\bar{n}=0$, which is long enough to ensure that $\bar{n}$ reaches equilibrium. We investigate how $\bar{n}$ changes for ten different TL thermometers, in the temperature range of 200-300 °C with 10 °C intervals, each having an independent set of kinetic parameters (see the supplement Table S1).

### 2.2.1 Linear thermal history

The thermal response of the dynamic equilibrium level of the trapped charge population ($\bar{n}$) of the ten thermometers is tested for three different linear thermal histories: 1) isothermal holding at 20 °C (Fig. 2a), 2) isothermal holding at 20 °C for 900 kyr

followed by linear cooling to 0 °C during the last 100 kyr (Fig. 2b), and 3) isothermal holding at 0 °C for 900 kyr followed by linear heating to 20 °C during the last 100 kyr (Fig. 2c). The results are shown in Fig. 2d, e and f respectively. In all cases, $\bar{n}$ is lowest for the lowest TL thermometers (or TL signals) and highest for the highest TL thermometers. For the isothermal scenario, $\bar{n}$ remains constant over the entire recent time (after reaching steady state) as there is no temperature change. For the cooling scenario, $\bar{n}$ increases as temperature decreases, because of a decrease of thermal loss. For the warming scenario, $\bar{n}$ decreases as temperature increases because of an increase of thermal loss. The increase and decrease of $\bar{n}$ (Figs. 2d and e) with temperature are most pronounced for lower temperature TL thermometers (<250 °C) and negligible for higher temperature TL thermometers (>250 °C). It must be noted that if the amplitude of temperature change (here 20 °C) were increased, the higher TL thermometers would change more dramatically. However, this would be unrealistic and beyond temperature variations observed at Earth's surface at comparable timescales.

*Figure 2*

The previous section shows that the thermal sensitivities of the different TL thermometers are distinct. It is therefore expected that their temporal sensitivities are also different. To quantify the temporal response of each TL thermometer, we prescribe a simple step function in which the temperature is set equal to 0 °C until a given time, $t_{change}$, and then increases to 10, 20 and 30 °C, in three different cases. We then calculate the present day trapped charge population ($\bar{n}_{present}$) for each TL thermometer. $t_{change}$ is varied from 100 kyr to the present.

The model predictions ($\bar{n}_{present}$) are shown as a function of time of change ($t_{change}$) in Fig. 3. In addition, we calculate the $t_{change}$ where $\bar{n}_{present}$ increases by 20 % compared to $\bar{n}_{present}$ predicted for a thermal history where $t_{change}$ is equal to 100 kyr. We define this corresponding time as the memory time ($t_{memory}$) at which the TL thermometer can record a temperature change. As expected, the lower TL thermometers record more recent temperature changes and the higher TL thermometer record older temperature changes. For example, in case of a 10 °C temperature change (Fig. 3a), the 200-210 °C thermometers record a change for ~10 kyr, while the 240-250 °C thermometers record a change for ~50 kyr. Furthermore, the number of sensitive thermometers increases as we raise the final temperature. For a 10 °C temperature change, only 5 TL thermometers (200-250 °C) record the temperature change (Fig. 3a), 6 TL thermometers (200-250 °C) record a 20 °C temperature change (Fig. 3b), and 7 TL thermometers (200-250 °C) record a 30 °C temperature change (Fig. 3c).

*Figure 3*

### 2.2.2 Periodic thermal history

Earth's climate varies in cyclical way, at multiple time scales from years to decades, centuries, and millennia, influenced by periodic variations in the Earth's orbit, known as Milankovitch cycles, at 25.77, 41 and 100 kyr. Therefore, temperatures are dictated by periodic functions that include several harmonics comprising decadal and millenial periods. In order to assess how the trapped charge population is affected by a periodic temperature history, we prescibe the following function as a thermal history.

$$T(t) = T_{mean} + T_{amp} \times \sin\left(\frac{2\pi}{P}t\right) \tag{3}$$

where $T_{mean}$, $T_{amp}$, and $P$ are the mean temperature, the amplitude, and the period of oscillation respectively.

Different combinations of arbitrary periodic thermal histories, with the same amplitude (10 °C), but three different periods (1, 10, 100 kyr) and three different mean temperatures (0, 15 and 30 °C) were used (solid lines of Fig. 4a,b and c) to assess the

effect of periodic temperature variations on $\bar{n}$ ($\bar{n}_{osc}$) for each TL thermometer (solid lines of Fig. 4d-l). These predictions are compared to isothermal effects on $\bar{n}$ ($\bar{n}_{iso}$) (dashed line in Fig. 4d-l) if the samples are kept at the mean temperature of the corresponding oscillation (dashed lines of Fig. 4a,b and c).

The results show that the $\bar{n}$ always depletes more for the lower temperature TL thermometer (e.g., 200-210 °C TL) than for the higher temperature TL thermometer (e.g., 290-300 °C TL), which results from a gradient in the thermal stabilities of lower to higher temperature TL thermometers. For every TL thermometer, $\bar{n}$ decreases if the mean temperature, $T_{mean}$, increases (Fig. 4d-f, g-i and j-l). This is because the probability of detrapping increases with increasing temperature. Finally, the higher temperature TL thermometers (near to 300 °C) remain relatively insensitive to such periodic temperature forcing ($T_{mean}$ up to 30 °C); with increasing $T_{mean}$ the higher temperature TL thermometers become more responsive.

*Figure  4*

One can also describe how the system behaves by comparing the period of oscillation (*P*) and the lifetime (or resident time) of trapped electrons ($\tau$) for a given temperature. For the 200 to 300 °C TL thermometers, $\tau$ spans ~10 kyr to 1 Gyr when the samples are at 0 °C; at higher temperatures $\tau$ is reduced as the probability of electron escape increases, reducing the lifetime to between ~0.1 kyr and 1 Myr when the samples are stored at 30 °C (Biswas et al., 2018). For P<< $\tau$ (e.g. Fig. 4d where P=1 kyr and $\tau$ spans ~10 kyr to 1 Gyr for the 200 to 300 °C TL thermometers for $T_{mean}$= 0 °C), the value of $\bar{n}_{osc}$ exhibits small fluctuations but always remains lower than $\bar{n}_{iso}$. This result implies that smaller periods (<1 kyr) and $T_{mean}$ (<30 °C) do not influence trapped charge equilibrium levels of 200 to 300 °C TL thermometers in an oscillating fashion and cannot be differentiated from the trapped charge population resulting from an isothermal condition. We must mention here, if the amplitude of oscillation increases the oscillating response to trapped charge equilibrium levels will be relatively prominent. However, the predicted values of $\bar{n}_{osc}$ are lower than those predicted using  constant temperature of $T_{mean}$, which is the mean temperature of the oscillation explored. Similarly, the present day $\bar{n}_{osc}$ remains indistinguishable from $\bar{n}_{iso}$ when P>>$\tau$ (e.g. see the behaviour of the low temperature TL thermometer shown in Fig. 4l). These two end-member scenarios are therefore not suitable for predicting the temporal variation of surface temperature. Interestingly, the response of $\bar{n}_{osc}$ deviates from its temperature forcing when $P\sim\tau$ (e.g., Fig. 4g-i and j-l). Under this condition, $\bar{n}_{osc}$ is out of phase and asymetric compared to the prescribed forcing, i.e., the thermal history. More importantly, the degree of deviation for different thermochronometers is different. Therefore, temperature variations can be reconstructed by targeting TL thermometers that have lifetimes of trapped electrons comparable to the period of surface temperature changes.

As discussed in the previous section, the response of trapped electron concentrations corresponding to a TL thermometer depends highly on the three characteristic parameters of the periodic forcing, i.e., $T_{mean}$, $T_{amp}$ and *P*. We now test the sensitivity of the model to these three parameters. The present day trapped charge population ($\bar{n}_{present}$) is predicted for different arbitrary combinations of $T_{mean}$ (0, 15 and 30 °C), $T_{amp}$ (5, 10 and 20 °C) and *P* (1, 10 and 100 kyr). The results show that $\bar{n}_{present}$ is highly dependent on the mean temperature variation, and less dependent on the amplitude and the period (Fig. 5). Although the $\bar{n}_{present}$ is less sensitive to the amplitude and the period, the pattern of $\bar{n}_{present}$ of different thermometers is unique. This ensures that complex thermal histories comprising mutiple harmonics with periods of about tens of kyr, but distinct from one another, can be recontructed.

*Figure 5*

## 3 Inversion of TL data into realistic thermal histories

The objective of this section is to test whether a temperature history can be recovered by inverting TL data into a realistic thermal history. We start the exercise by predicting TL data using Eq. 1 for a specific thermal history (forward modeling) that we then invert using a Bayesian approach (Biswas et al., 2018; King et al., 2016a; King et al., 2016b) (inverse modeling). This synthetic experiment is performed in two different cases. First, we describe the forward and inverse modeling in a general way and then report the two synthetic examples.

### 3.1 Forward modeling

Forward modelling is achieved by solving Eq. 1 and prescribing a thermal history, similarly to the previous sections. This approach enables us to predict the present day trapped charge population for a specific TL thermometer using the kinetic parameters extracted for sample MBTP9. Using this approach, we generate a range of "observed values" ($\bar{n}_{obs}$) for a particular thermal history, which we then try to recover using an inversion method. We run the model for 1 Ma to ensure that $\bar{n}$ reaches steady state assuming an initial condition of $\bar{n}=0$. For a specific thermal history, TL thermometers with lower thermal stability exhibit lower $\bar{n}_{obs}$ than TL thermometers with higher thermal stabilities. It is worth noting that the predicted $\bar{n}_{obs}$ values are mostly sensitive to two parameters, the trap depth ($E$) and athermal fading ($\rho'$), such that these must be constrained carefully from laboratory experiments.

### 3.2 Inverse modeling

To invert TL data ($\bar{n}_{obs}$) into a thermal history, a Bayesian approach is used. We first generate a large number of random thermal histories (300,000). For each random path, the present-day TL signals are predicted by solving Eq. 1. Each predicted present day TL values ($\bar{n}_{predict}$) is then compared with the observed TL ($\bar{n}_{obs}$) using the following misfit function (Wheelock et al., 2015) and likelihood:

$$misfit = \frac{1}{l}\sum_{i=1}^{l}\left[\frac{1}{2}\times\frac{\bar{n}_{obs}}{\sigma_{\bar{n}_{obs}}}\times\log\frac{\bar{n}_{predict}}{\bar{n}_{obs}}\right]^2 \tag{4}$$

$$likelihood = \exp(-misfit) \tag{5}$$

where $l$ is the number of TL thermometers (here $l$=4) and $\sigma_{\bar{n}_{obs}}$ is the uncertainty of corresponding $\bar{n}_{obs}$. An arbitrary uncertainty of 20% of $\bar{n}_{obs}$ is assumed for synthetic test. For each random path $\bar{n}_{predict}$ for a specific thermometer is usually calculated using mean values of the specific set of kinetic parameters (Biswas et al., 2018; King et al., 2016a; King et al., 2016b). However, the kinetic parameters have uncertainties as shown in Fig. 1 and supplementary Table S1. To accommodate the measurement uncertainties in kinetic parameters, for each random thermal history, we randomly picked the kinetic parameters within its error range. Finally, $\bar{n}_{obs}$ was also randomly picked within its error range assuming that any value within error limit is equally probable (c.f. Guralnik et al., 2015). Since the randomization is applied to a large number of parameters, it is necessary to run the model for large number of thermal histories (300,000 iterations here).

The thermal histories that best fit the data are selected using a rejection algorithm that satisfies the criterion *likelihood* >R, where $R$ is a random number between 0 and 1. A probability density distribution is then constructed by counting the number of accepted thermal histories passing through each grid cell, which is generated by dividing the time-temperature space (0-100 ka and -50 to 50 °C) into 100×100 cells. This approach is commonly used in different thermochronometric studies (Biswas et al., 2018; Braun et al., 2012; Gallagher et al., 2009; King et al., 2016b). It should be noted that the misfit function (Eq. 4) used here is different to the one used in previous studies (Biswas et al., 2018; King et al., 2016b) but is the same as that used in

King et al. (2019). We find that a log misfit enables us to better fit data that vary across orders of magnitude, as trapped charge populations vary greatly for different TL signals.

### 3.3 Synthetic approach 1

We choose three arbitrary periodic thermal histories, Path1 ($T_{mean}$= 10 °C, $T_{amp}$= 10 °C and $P$= 25.77 kyr), Path2 ($T_{mean}$= 20 °C, $T_{amp}$= 10 °C and $P$= 25.77 kyr) and Path3 ($T_{mean}$= 10 °C, $T_{amp}$= 20 °C and $P$= 25.77 kyr) as shown Fig. 6a. For each thermal history, the present day trapped charge concentrations ($\bar{n}_{obs}$) are calculated for four TL thermometers (210-250 °C, 10 °C interval) as described in section 3.1 and represented in Fig. 6b-d. We then invert the TL data ($\bar{n}_{obs}$) into a thermal history as described in section 3.2. For the inverse modelling, we first generate a large number of random periodic histories (300,000) with $T_{mean}$ and $T_{amp}$ randomly varying from 0 to 50 °C, $P$ randomly varies between three cycles, 25.77, 41 and 100 kyr. We do not vary $P$ in a completely random fashion because $\bar{n}_{obs}$ is less sensitive to $P$; i.e., it is difficult to resolve neighboring periods (as discussed in section 2.2.2 and Fig. 5). The results are shown in Fig. 6e-j. Although this approach predicts the very recent temperature well (up to max 5 ka) it loses the periodic information (25.77 kyr) because of the significant number of accepted thermal histories with different periods (41 and 100 kyr). The same exercise was repeated but fixing the period to 25.77 kyr for the inversion. The results are shown in Fig. 6k-p. Interestingly, this approach enables to recover the actual solution within $1\sigma$ uncertainty. This shows that a periodic thermal history can be predicted well if the period is known a priori; it enables to constrains $T_{mean}$ and $T_{amp}$ satisfactorily. To circumvent the limitation (period) of this method we use synthetic approach 2, the $\delta^{18}$O data to impose the shape of the thermal histories as a priori information. This is typically done for inversion problem when appropriate. We then constrain $T_{mean}$ and $T_{amp}$ of the spectrum (as discussed in next section).

*Figure 6*

### 3.4 Synthetic approach 2

Here we report the result of three thermal histories assuming that the temperature follows the measured $\delta^{18}$O records from Greenland for the past 60 ka, which is based on various records from the DYE-3, the GRIP, and the NorthGRIP ice cores (Svensson et al., 2008). For our purpose, we scale the $\delta^{18}$O records to thermal histories and assume a constant temperature prior to 60 ka (Fig. 7a). For each thermal history, the present day trapped charge concentrations ($\bar{n}_{obs}$) are calculated for four TL thermometer (210-250 °C, 10 °C interval). The results reported in Fig. 7b-d show a clear depletion of the trapped charge population during the last 20 kyr, for all investigated scenarios. However, high frequency temperature variations are dampened, implying that TL thermometry is insensitive to short term variations. This result is consistent with the results shown in Fig. 4. For the inverse modelling, we first generate a large number of random periodic histories (300,000) assuming that they all follow the Greenland ice core $\delta^{18}$O record (Svensson et al., 2008), which we scaled randomly by varying the amplitude of the temperature oscillation (i.e., the difference between minimum the temperature, which is at ~20 ka, and the maximum temperature at present) between 0 and 40 °C and the minimum temperature (i.e., temperature at ~20 ka) from -20 to 30 °C (see supplement S2). Note that making this assumption is somewhat equivalent to assuming a prior estimated on the inferred thermal history (Tarantola, 2005). The inversion results for three tested thermal histories are shown in Fig. 7e-p. The probability density functions (Fig. 7e, f and g) show it is possible to recover all three thermal histories within the $1\sigma$ confidence level using this inversion approach.

*Figure 7*

## 4 Proof of concept

The following sections explore the potential of multi TL thermometers in the lower temperature region of the TL glow curve (210-250 °C) to infer the rock temperature histories for two samples collected in the European Alps.

### 4.1 Sample location

Two bedrock samples (MBTP1 and MBTP9) were collected at the Mer de Glace glacier (Mont Blanc massif, European Alps) at an altitude of 2545 and 2133 m. The rock surfaces were exposed since the last glacial maximum (LGM); with exposure ages younger than the LGM of about 20 kyr, based on [10]Be terrestrial cosmogenic nuclide and OSL surface exposure dating (Lehmann et al., 2019; Lehmann et al., 2020).

### 4.2 Sample preparation

The sample preparation followed the method reported previously (King et al., 2016b). The light exposed outer layer (>2 cm from the surface) was removed using a diamond saw under subdued red-light conditions with constant water flow to avoid frictional heating. The interior part of the sample was gently crushed with a mortar and pestle and sieved to separate the 150–250 μm grain size. The samples were sequentially treated with 10% HCl and 30% $H_2O_2$ to remove carbonate and organic matter respectively. Once dried, the magnetic fractions were removed using a hand magnet. The K-feldspar fraction was separated by density separation (<2.58 gm/cm$^3$) using sodium polytungstate. The grains were mounted on stainless steel discs using Silko-spay. Small aliquots of 2 mm diameter (containing ~100 grains) were prepared as these feldspars were highly luminescent.

### 4.3 Experimental procedure

The TL luminescence measurements were made using a Risø TL/OSL reader (TL/OSL DA- 20; Bøtter-Jensen et al., 2010) equipped with a $^{90}Sr/^{90}Y$ irradiation source (~ 0.24 Gy/s) at the University of Lausanne. A heating rate of 1 °C/s was used, under constant flow of $N_2$ gas. The TL emission was restricted to violet-blue (395±30 nm) using a filter combination of BG3 and BG39. The measurement details are discussed below. Typically, the minimum detectable limit for the present instrument is ~ 300 photon counts per second (cps) considering the signal should be three times of background level which is ~ 100 cps. The present high luminescent feldspar has maximum photon count of ~10$^6$ cps. This restricts to use the TL signals up to ~10$^{-3}$ % of maximum TL signals.

#### 4.3.1 Measurements

Following Biswas et al. (2018), three sets of experiments were performed to constrain the growth parameters ($D_0$, $a$), thermal decay parameters ($E$, $s$, $b$) and athermal decay parameters ($\rho'$). The athermal frequency factor ($\tilde{s}$) is taken as $3 \times 10^{15}$ s$^{-1}$ (Huntley, 2006).

The growth parameters and the natural TL level, i.e., the trapped charge population ($\bar{n}_{obs}$), are estimated using the multiple aliquot regeneration dose (MAR) protocol (Aitken, 1985) with post-glow normalization (Tang and Li, 2017). Eight regeneration doses (0, 24, 47, 118, 236, 472, 944 and 1888 Gy) were given and three aliquots were used for each dose point. A cut-heat of 200 °C was applied to remove traps that are unstable over laboratory timescales. We observed a significant sensitivity change (decrease) during the very first measurement of natural TL, which means that natural and regenerative TL signals were not measured under identical TL sensitivity conditions. To circumvent this sensitivity change, we adopted the

natural correction factor method (NCF; Chauhan and Singhvi, 2019; Singhvi et al., 2010; Singhvi et al., 2011). However, the NCF was initially developed for quartz OSL (Singhvi et al., 2011), it should to be adapted for feldspar.

The NCF method for quartz relies on the fact that the 110 °C TL peak and the blue stimulated OSL are correlated (Singhvi et al., 2011). In contrast, TL of feldspar does not exhibit a distinct 110 °C TL peak. The luminescence process in feldspar is more complicated because it arises from a continuous distribution of trapping energies and the dose response characteristics ($D_0$) varies along TL glow curve (Fig. 1). To circumvent these issues, we proceeded as follows. We first give a small dose (i.e., <100 Gy) in addition to the natural dose and subsequently measure the TL signal up to 200 °C ($TL_1$). Then the sample is annealed by heating it to 450 °C, which is followed by a dose of the same amount and measurement of the TL glow curve to 200 °C ($TL_2$). We observe that the TL sensitivity of the natural measurement is higher than the post natural regeneration measurement (Fig. 8a). We then calculate the NCF at different temperature between 90 to 150 °C, similar to the use of the 110 °C TL peak for quartz. We find that the NCF decreases with increasing temperature during the TL measurement (Fig. 8c). Since there is no direct way to measure the NCF beyond 150 °C, we then extrapolate the NCF value at the region of interest to higher temperatures, i.e., 210-250 °C (Fig. 8c), which we call var-NCF. In turn, the trapped charge population ($\bar{n}_{obs}$) is corrected with the corresponding factor which is in between 1 and 2 for sample MBTP1 and MBTP9 (Table 1 and Fig. 8). Finally, we investigated the effects of variable doses and the NCF and found it had no effect for doses below 100 Gy (Fig. 8b).

*Figure 8*

The thermal decay parameters are estimated using the $T_m$–$T_{stop}$ method (McKeever, 1980) and analyzed by subtraction and fitting of sub-peaks (Pagonis et al., 2014). For the athermal decay parameter, a fading experiment (Huntley and Lian, 2006) was performed for different delay times; aliquots were preheated to 200 °C prior to storage.

**4.3.2 Estimating the kinetic parameters**

The kinetic parameters of growth ($D_0$, $a$), thermal decay ($E$, $s$, $b$) and athermal decay ($\rho'$) were inferred using the approach of Biswas et al. (2018) for all thermometers (210-250 °C, 10 °C interval). The results are summarized in Table 1 and shown in supplementary material S3. It can be noted that with increasing the TL temperature (or thermometer), the activation energy ($E$) increase and athermal fading ($\rho'$) decreases (Table 1). The dose rate ($\dot{D}$) values, another growth parameter, were taken from Lehmann et al. (2020). Since a cut-heat of 200 °C was applied for the MAR growth analysis and fading experiments, we focus on the 210-250 °C TL thermometers (i.e., four thermometers). We did not use TL signals beyond 250 °C, as they are insensitive to typical surface temperature fluctuations, as discussed in section 2.2.

*Table 1*

**4.4 Predicting the surface temperature**

The measured TL signals ($\bar{n}_{obs}$) are then inverted to infer the thermal history as described in section 3.2 and 3.4 (Synthetic approach 2). For the thermal histories, we again use the Greenland ice core $\delta^{18}O$ record (Svensson et al., 2008), which we scaled as described in section 3.2. We assume that the atmospheric temperatures of the Mont Blanc massif followed the trend observed for the Greenland ice core data over the last 60 ka. Note that temperature increase during the last glacial cycle was synchronous with the temperature anomalies observed in Greenland (e.g., Heiri et al., 2014; Schwander et al., 2000; van Raden et al. 2013). The rationale here is that all temperatures in the Mont Blanc massif follow the Greenland ice core $\delta^{18}O$ data but the amplitude of temperature oscillation (minimum temperature at ~20 ka to maximum temperature at the present day) and mean temperature are unknown. We pick the amplitude of temperature oscillation randomly between 0 and 40 °C, and the

base temperature (temperature at ~20 ka) between -20 and 30 °C. By generating a large number of random thermal histories (300,000), the probability density function is constructed as discussed in section 3.2. The results of two samples, MBTP1 and MBTP2, are shown in Fig. 9 and suggest that the temperature rose from $-4.6^{+3.7}_{-4.1}$ to $6.2^{+3.1}_{-3.5}$ °C for sample MBTP1 and $-2.0^{+3.9}_{-4.1}$ to $7.9^{+3.0}_{-3.1}$ °C for sample MBTP9, since 20 ka, considering one sigma uncertainty. The inferred median suggests an increase of ~10-11 °C for the rock surface temperature over the last 20 ka.

*Figure 9*

## 5 Discussion

The theoretical model for the rate equation of trapped charge population in feldspar has been described in several ways; first order kinetics (Brown and Rhodes, 2017; Yukihara et al., 2018), general order kinetics (Biswas et al., 2018; Guralnik et al., 2015b), charge transport through sub-conduction band-tail states (King et al., 2016a; Li and Li, 2013), Gausian distribution of trapped energies (Lambert et al., In Revision), or localized recombination in randomly distributed defects (Jain et al., 2012). What is common to all these models is that luminescence of feldspar is complicated and exhibit a non-linear non-first order kind of behaviour due to either presence of sub-conduction band-tail states (Morthekai et al., 2019; Poolton et al., 2002) or complex charge transport mechanism. TL in feldspar is even more complicated because it shows continuous distribution of trapping energies (Biswas et al., 2018; Duller, 1997; Grün and Packman, 1994; Pagonis et al., 2014; Strickertsson, 1985) and TL is a more diffusive process than OSL; OSL of feldspar has resonant energy levels (Hütt et al., 1988). Different models were reviewed and tested by Guralnik et al. (2015b) who suggested that the general order kinetic, a mathematically simplified model, could be used to explain luminescence phenomenon well. We adopted this model for TL of feldspar where the power terms (*a*, *b*) accounts for the nonlinearity involved in the TL of feldspar. The efficacy of using general order kinetics has been demonstrated to samples with known thermal history (KTB borehole samples) for OSL of feldspar (Guralnik et al., 2015a) and TL of feldspar (Biswas et al., 2018).

Here we investigate the difference in temperature sensitivity of different TL thermometers, which correspond to individual TL temperature or TL signals. On the basis of the kinetic parameters derived for our sample, and our sensitivity tests (section 2.2.1), we recommend using TL thermometers with temperature range of 200 to 250 °C for a typical surface temperature fluctuation, e.g. ~10 °C. If the temperature fluctuations are larger, higher temperature TL (>250 °C TL) can be used. The multiple TL signal (200 to 250 °C, 10 °C interval) can constrain thermal history of ~50 kyr. A higher temperature fluctuation can be better constrained with a greater number of thermometers (as discussed in section 2.2.1).

For periodic oscillations, when the period is comparable to the lifetime of the trapped electron for a given thermometer, it may be used to infer temporal variation of surface temperature (see section 2.2.2). Typically, tens of kyr of temperature oscillation can be detected using TL thermometers with peak temperatures higher than 200 °C (210 to 250 °C). Periodic oscillation with lower period (<1 kyr) will exhibit a similar effect to isothermal temperature condition, yielding a temperature higher than the mean of oscillation.

One outstanding issue when using TL is the sensitivity change during the very first measurements up to 450 °C, which cannot be corrected by post-glow normalization (Tang and Li, 2017). Here we show that sensitivity changes during natural measurements can be monitored for lower temperature TL (<150 °C) following the same method adopted for the OSL of quartz, which is called the natural correction factor (NCF; Singhvi et al., 2011). Because there is no direct method to track the TL sensitivity change in the region of interest (210-250 °C TL), we simply extrapolate the sensitivity change observed in the lower temperature TL peaks (i.e., 90-150 °C) to the region of interest (i.e., 210-250 °C). This is new and it will need further

investigation. However, we find that the effect of the initial sensitivity change on the amplitude of the inferred temperature histories is small. In Fig. 10, we compare inversion results for three different scenarios for sample MBTP1; 1) there is no initial sensitivity correction, i.e., NCF =1; 2) the initial sensitivity correction is done using the value obtained at 100 °C ($NCF_{100}$=1.64±0.08); and 3) the var-NCF approach described in the section 4.3.1 is used. Although the results show that the sensitivity correction has a significant impact on the absolute inferred temperature, we do not observe much difference between using a constant value and the extrapolated value. Furthermore, and more importantly, the difference between the present-day temperature and temperature about 20 ka remains about 10-11°C in the three tested cases (median of the prediction).

*Figure 10*

The estimated constant erosion rates in these two sample locations, MBTP1 and MBTP9, are $3.5\times10^{-3}$ and $3.2\times10^{-2}$ mm/yr with maximum possible time of erosion of 20.9 and 19.5 ka respectively (Lehman et al. 2020). This translates to a maximum erosion depth in these two locations are 0.07 and 0.62 m respectively. At those depths, mean temperature should be in equilibrium with atmospheric temperature (e.g., Hasler et al. 2011). Based on the rock temperature measurement of borehole samples in Mont-Blanc massif, Magnin et al. (2017) suggest up to a 2 m depth the rock temperature is nearly constant with depth or a maximum variation of up to 1 °C (Fig. 6c of Magnin et al., 2017). No effect of erosion on surface temperature is considered here.

For the inverse modeling of natural samples, $\delta^{18}$O data are used as a prior on the shape of the thermal histories, but we leave two scaling parameters free – minimum temperature at 20 ka, and amplitude (temperature difference between at 20 ka and present)– and we did not include the role of ice on setting the rock temperature during glaciation. Lehman et al. (2020) provided a range of solution for deglaciation in the present location, either thinning of 450 m glacier occurred progressively between ~17 ka and ~12 ka or it was instantaneous. In absence of clear scenario, we took two samples from two extreme altitudes of 2545 (MBTP1) and 2133 (MBTP9). The top most sample, MBTP1, had a very thin or no glacier covered during LGM, and the bottom most sample, MBTP9, was exposed or covered by ice. Thus, it is expected that the during LGM, rock temperature of MBTP1 would have been in equilibrium with atmospheric temperature (Hoelzle et al. 1999) whereas it is less clear for MBTP9. If it was covered by ice, it was likely temperate ice and  the basal temperature would be close to 0 °C. Interestingly, we find similar results from the inversion; predicted rock surface temperatures of MBTP1 and MBTP9, during LGM, are $-4.6^{+3.7}_{-4.1}$ and $-2.0^{+3.9}_{-4.1}$ respectively; temperature of the bottom most sample during LGM is close to 0 °C, at least within error.

The application of the introduced method predicts that the final rock surface temperatures at the locations of Mont-Blanc massif are 6.2 ± 3.2 °C for MBTP1 (2545 m) and 7.9 ± 3.0 °C for sample MBTP9 (2133). Although these temperatures have large uncertainty, they are higher than the mean annual atmospheric temperature in this location. The mean annual temperature of Chamonix (1035 m), a nearby city, is 7.3 °C. Considering an adiabatic lapse rate of 5 °C/km, the expected mean annual atmospheric temperatures at the sample location of MBTP1 and MBTP9 are ~0 and 2 °C respectively. The offset between the predicted and expected temperature (~6 °C) can be explained by two main reasons: 1) the rock surface temperature is always higher than atmospheric temperature and the temperature difference can be up to 10 °C (Magnin et al., 2019), and 2) seasonal temperature fluctuations may lead to an overestimation of the mean annual temperature (as discussed in section 2.2.2). To quantify this latter offset, we performed a simple synthetic test, with annual oscillation of +10 °C (summer) to -10°C (winter) with mean at 0 °C, up to 20 ka (before that temperature was set to a 0 °C isotherm), and predicted the equivalent isothermal temperature using the inverse approach. The result suggests a mean annual temperature that is 2.7±0.7 °C higher than the mean temperature of the periodic signal (Fig. 11), confirming the results of Guralnik and Sohbati (2019).

*Figure 11*

The inverse modeling results show an increase of rock surface temperature in the Mont Blanc Massif of ~10-11 °C (considering median of the prediction) from 20 ka to today. The median of the distribution of possible thermal histories of the two samples follows Greenland ice core $\delta^{18}O$ anomalies with missing of low frequencies. Climate reconstructions in Europe using fossil pollen suggest that the mean annual temperature anomaly (the difference between the temperature at the LGM and today) is 12±3 °C in the north of Pyrenees–Alps line (Peyron et al., 1998). Wu et al. (2007) inferred that LGM temperatures in Europe were ~ 10-15 °C lower than the present-day temperature based on pollen analysis. Although there are large uncertainties associated with pollen data and having its methodological constrain, the overlap in temperature estimates between the two proxies suggest that TL may be a reliable paleothermometer.

## 6 Conclusions

A new approach to reconstruct the temporal variation of rock surface temperature using the TL of feldspar is introduced. Forward modeling of different TL signals suggests that TL signals in the range of 210 to 250 °C are sensitive to typical surface temperature fluctuations, which we define as TL thermometers. Multiple TL thermometers (210-250 °C, 10 °C interval) can then be used to constrain thermal histories of rocks over ~50 kyr for temperature fluctuations of ~10 °C. The sensitivity of the periodic forcing on trapped charge populations suggest that natural TL is sensitive enough to mean temperature and amplitude of periodic forcings. Typically, tens of kyr of temperature oscillation can be predicted using this approach. Finally, we show that it is possible to recover thermal histories of rocks when one assumes that the temperature followed observed Greenland ice core $\delta^{18}O$ record.

## Code/data availability

The raw data files and Matlab codes can be obtained by requirement to the corresponding author.

## Author contribution

RHB and FH conceived the idea. RHB, FH and GEK conceptualized the study. RHB designed the experiments and numerical modelling with inputs from FH. BL collected the samples and looked into the Geological aspects of the location. Sample preparation, measurement and analysis were made by RHB with inputs from AKS. RHB wrote the paper with input from all co-authors.

## Competing interests

The authors declare that they have no conflict of interest.

## Acknowledgements

RHB acknowledge University of Lausanne for support and Prof. Jean Braun for insightful discussions. We thank Benny Guralnik for his astute opinion on this work. Florence Magnin is thanked to help us understand the present rock temperature scenarios. AKS thanks the Indian Department of Science and Technology, SERB- Year of Science Chair Professorship.

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

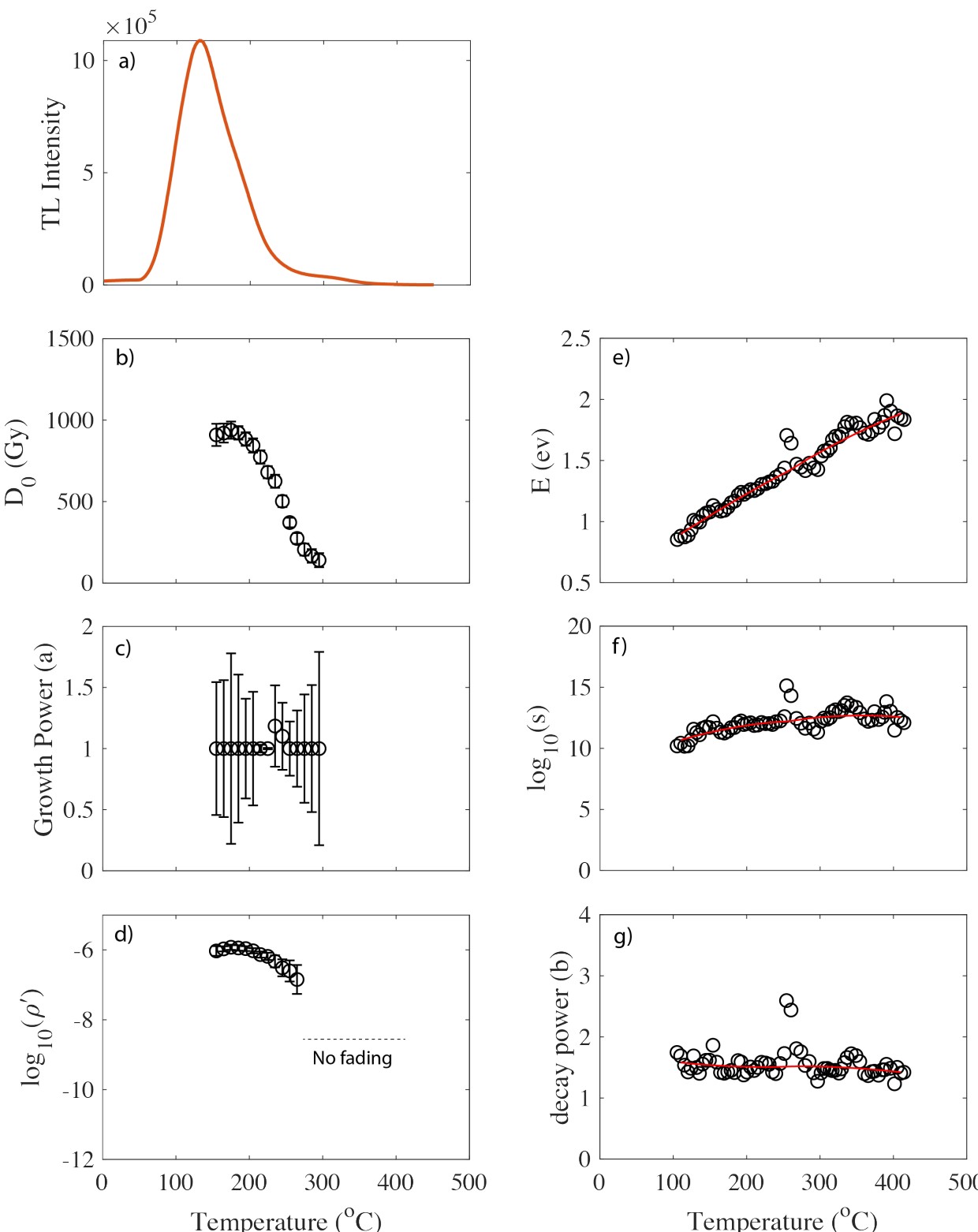

Figure 1: Inferred TL kinetic parameters from TL glow curve of sample MBTP9. The method used to constrain these parameters in the laboratory is fully explained in Biswas et al. (2018) and described in supplementary material S1.

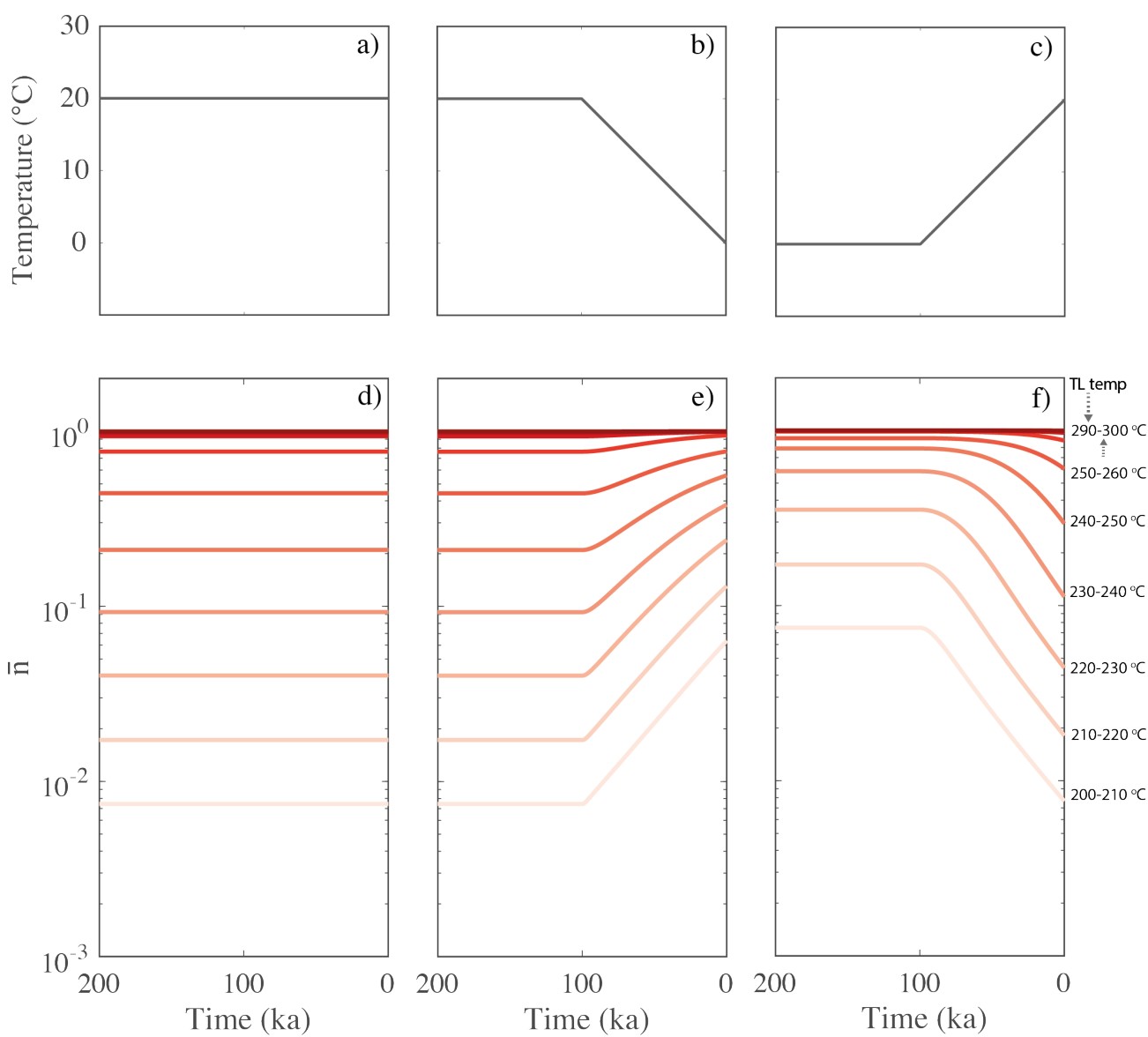

**Figure 2: a, b, and c are the three prescribed thermal histories: isothermal, cooling and warming respectively. d, e and f are the corresponding dynamic equilibrium levels of trapped charge population ($\bar{n}$) of 10 different thermometers in the range of 200-300 °C. The temperature of the thermometers (or TL temperature) are shown on the right.**

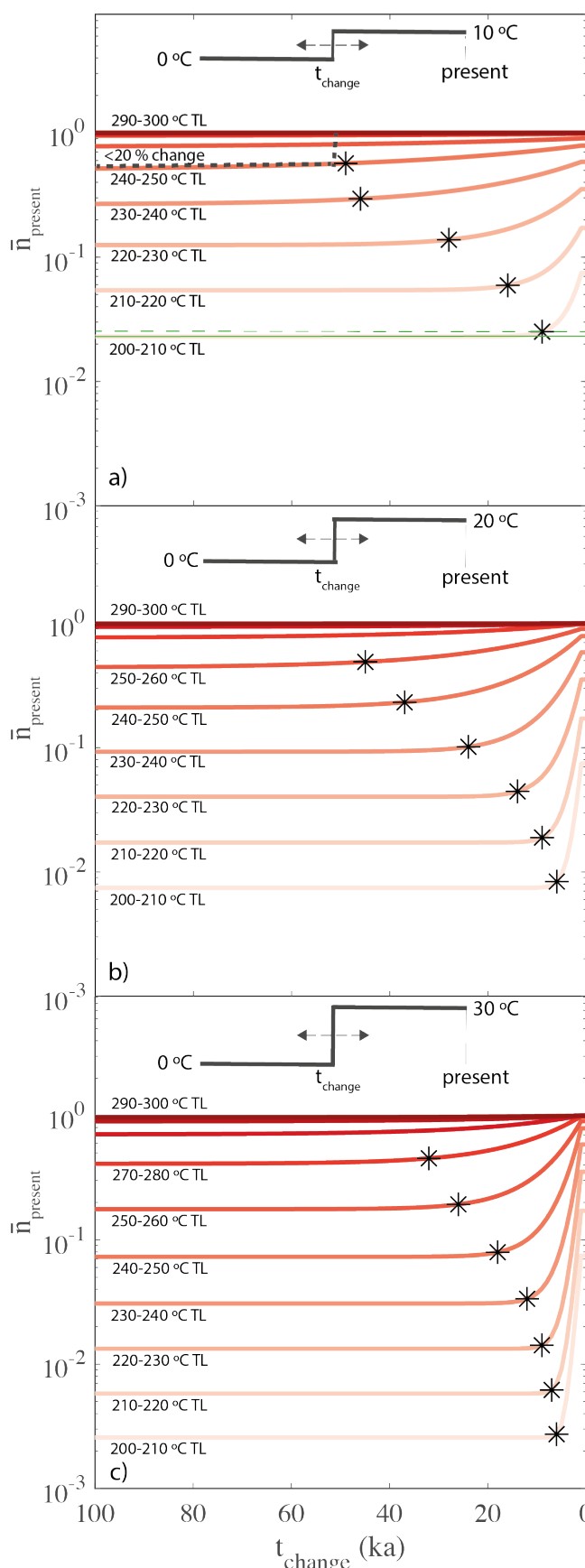

**Figure 3: The variation of present day trapped charge population ($\overline{n}_{present}$) of different thermometers with time of change of temperature ($t_{change}$) of step function like thermal history with temperature change of a) 10 °C, b) 20 °C and c) 30 °C. The asterisk symbols denote the memory time ($t_{memory}$) that a thermometer can record the temperature change history. Estimation of $t_{memory}$ is illustrated in plot a for 200-210 °C TL thermometer. The solid green line corresponds to $\overline{n}_{present}$ for $t_{change}$ =100 ka, and the dashed green lines represent $\overline{n}_{present}$ for $t_{change} = t_{memory}$, which is 20% higher than the solid green line.**

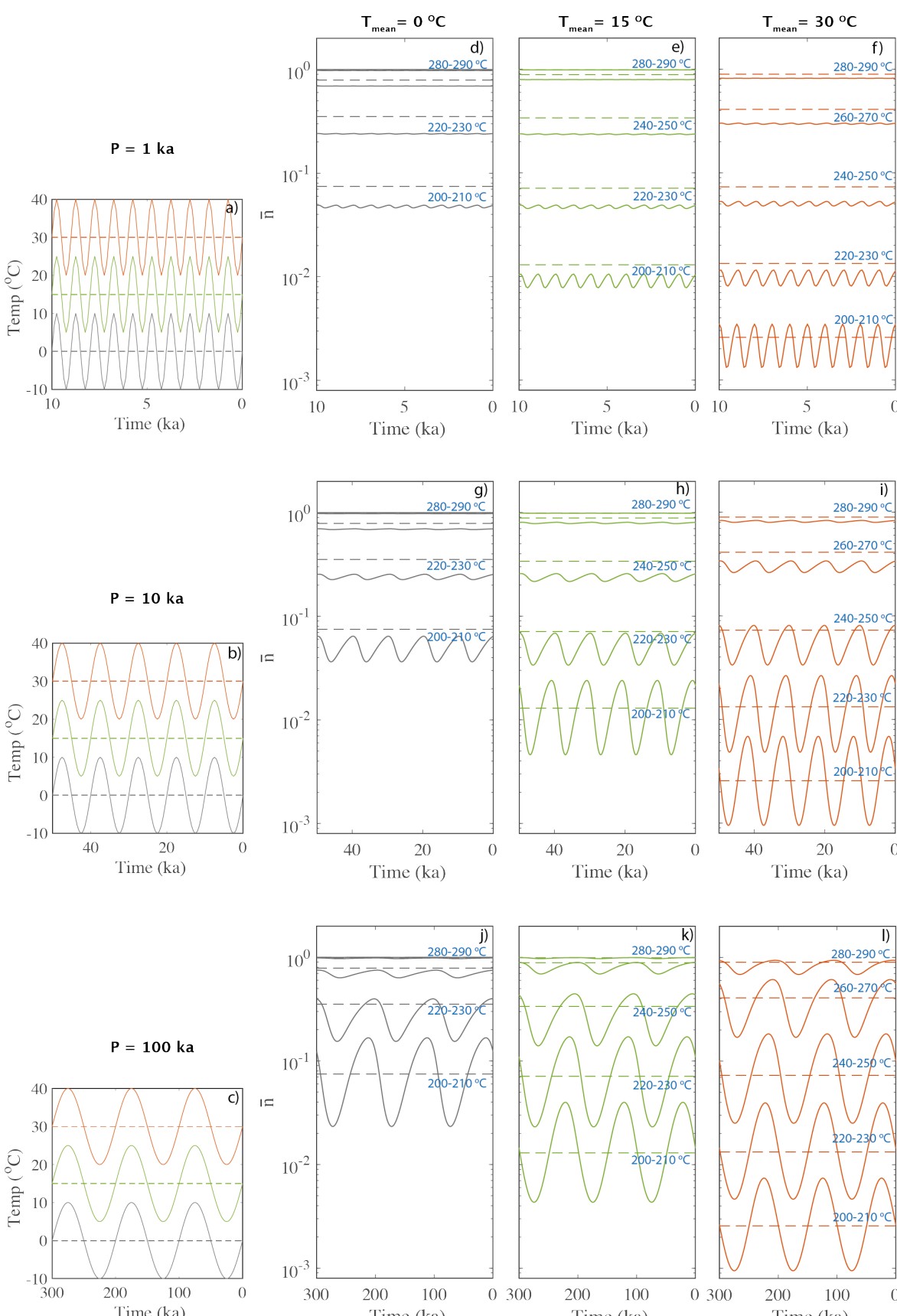

**Figure 4: Variation of trapped charge population ($\overline{n}$) for different sinusoidal thermal histories. a, b, and c represents the prescribed thermal histories with same mean amplitude (10 °C) and three different mean temperatures (0, 15 and 30 °C) but for different periods, 1, 10 and 100 ka respectively. The dashed lines are the isotherm (mean temperature of oscillation). Fig. d-f, g-i, and j-l are the the response of $\overline{n}$ for the corresponding thermal field. The solid lines are for oscillating fields and dashed lines are for isothermal fields. The temperature of the representative thermometers (or TL temperature) are shown inside in blue.**

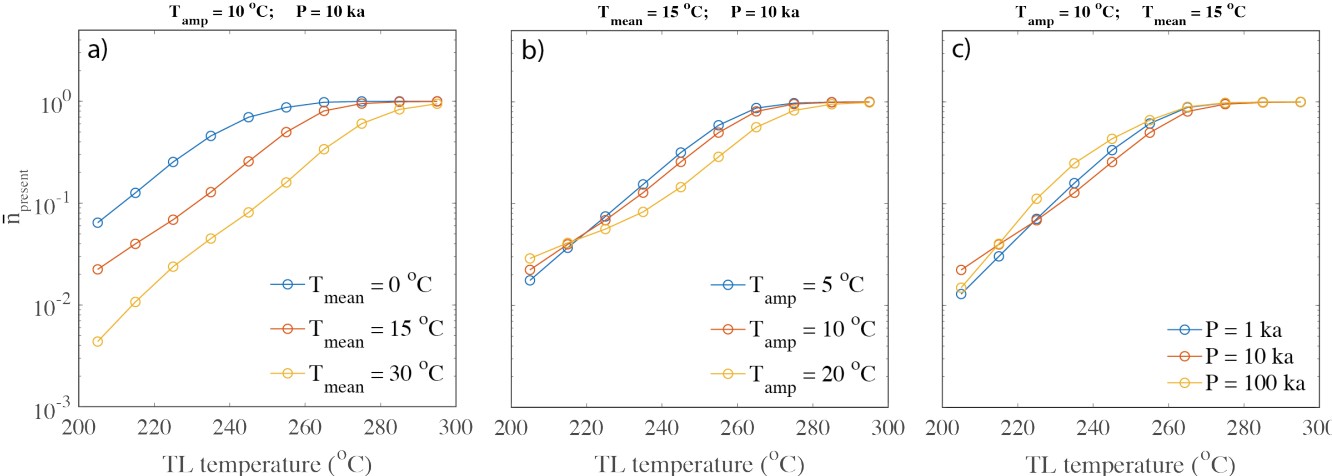

**Figure 5: Present day trapped charge populations ($\overline{n}_{present}$) of different TL signals (or thermometers) with a) mean temperature variation (amplitude and period are fixed), b) amplitude variation (mean temperature and period are fixed), and c) period variation (mean temperature and amplitudes are fixed).**

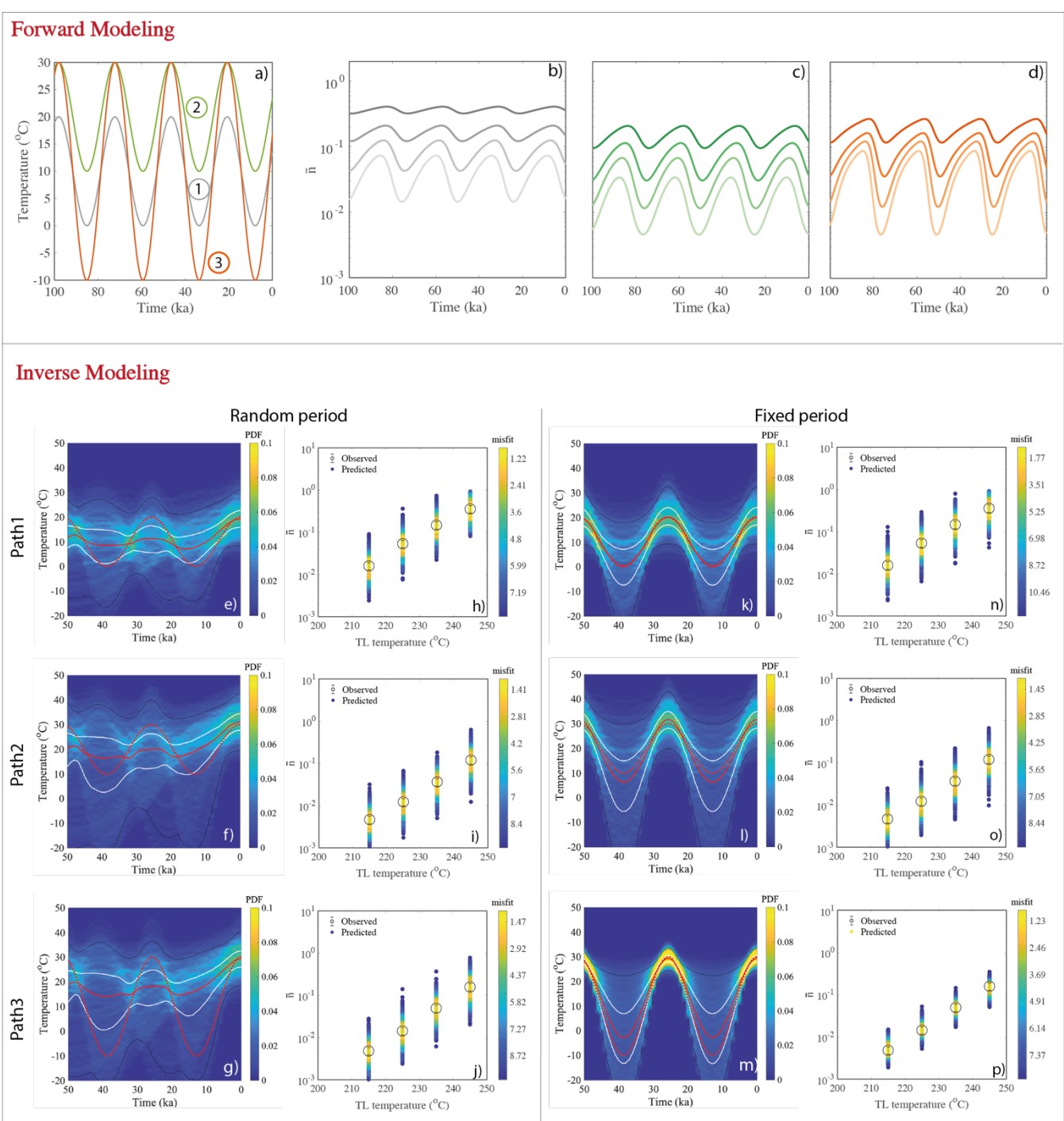

Figure 6: Results of synthetic experiment of approach 1, a-d) are for the forward modeling and e-p) are for the inverse modeling. a) three arbitrary periodic thermal histories with different mean temperature ($T_{mean}$) and amplitude ($T_{amp}$) but fixed period $P$. b), c) and d) are the evolution of trapped charge population ($\bar{n}$) for four different thermometers (210-250 °C, 10 °C interval) of the corresponding to three thermal histories, respectively. The present day trapped charge populations are considered as observed values ($\bar{n}_{obs}$) for the inverse modeling. e), f) and g) are the inferred probability density plots when $T_{mean}$, $T_{amp}$ and $P$ are randomly varied. h), i), and j) depict the fit between the observed TL (obtained through forward modeling). The solid red lines show the predicted median, white lines and black lines show the $1\sigma$ and $2\sigma$ confidence intervals in the probability density distribution k), l) and m) are the inferred probability density functions when $T_{mean}$, and $T_{amp}$ are randomly varied but $P$ is fixed. n), o), and p) depict the fit between the observed TL (obtained through forward modeling). These results are obtained using the kinetic parameters of sample MBTP9.

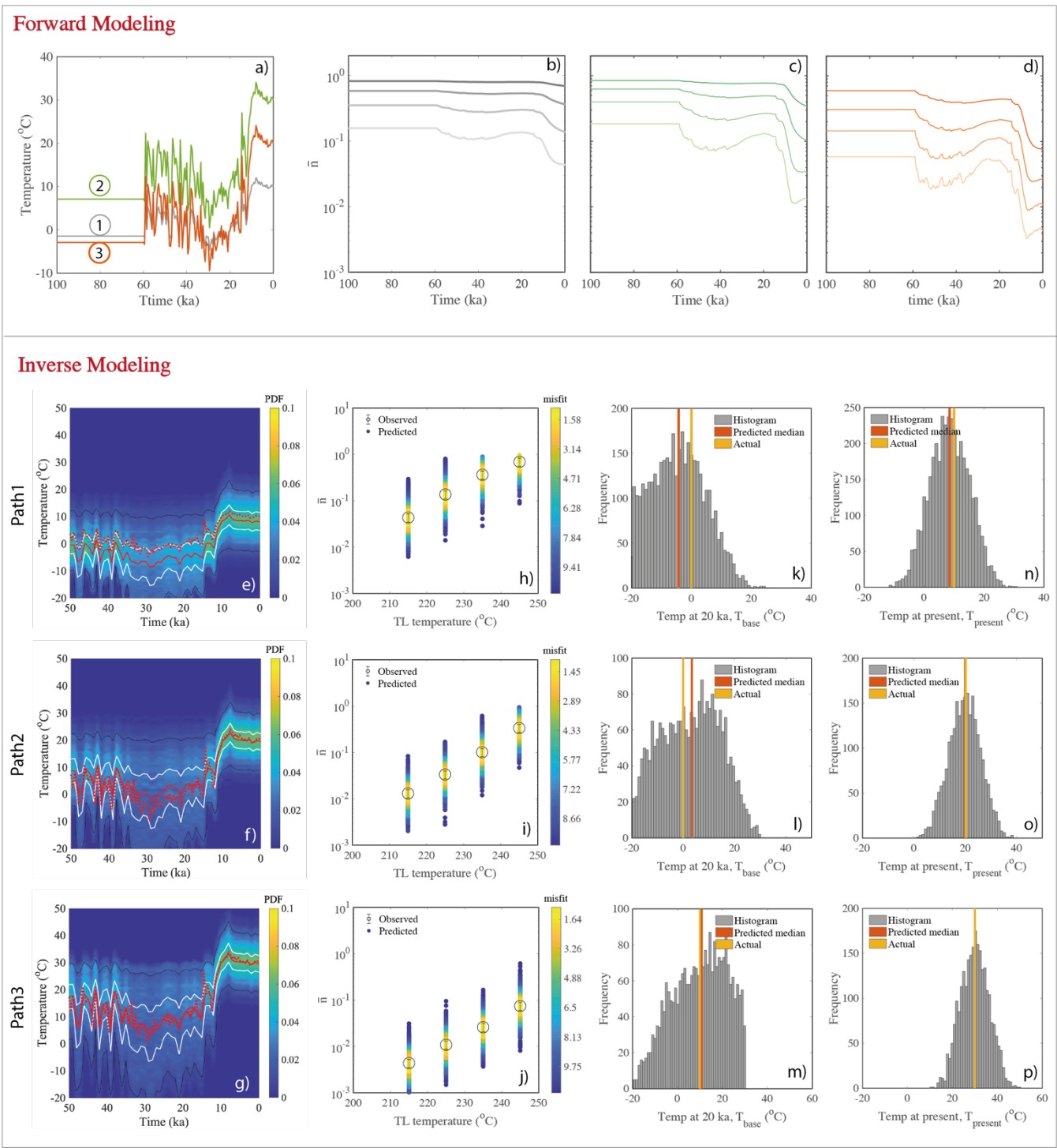

**Figure 7: Results of synthetic experiment of approach 2, a-d) are for the forward modeling and e-p) are for the inverse modeling. a) three arbitrary temperature histories obtained by scaling the Greenland δ¹⁸O ice core record (Svensson et al., 2008). b), c) and d) are the evolution of trapped charge population ($\bar{n}$) for four different thermometers (210-250 °C, 10 °C interval) corresponding to three histories respectively. The present day trapped charge populations are considered as observed value ($\bar{n}_{obs}$) for the inverse modeling. e), f) and g) are the inferred probability density functions. The dashed red lines show the actual temperature history as used in forward modeling, the solid red lines show the predicted median, white lines and black lines show the 1σ and 2σ confidence intervals in the probability density distribution. h), i), and j) depict the fit between the observed TL (obtained through forward modeling) and modelled TL (obtained through inverse modeling). k), l), and m) represent histograms of the parameter, base temperature, $T_{base}$ (which is temperature at 20 ka). n), o), and p) represent histograms of present temperature, $T_{present}$ (which is $T_{base}+T_{amp}$ as shown in supplement Eq. S9). These results are obtained using the kinetic parameters of sample MBTP9.**

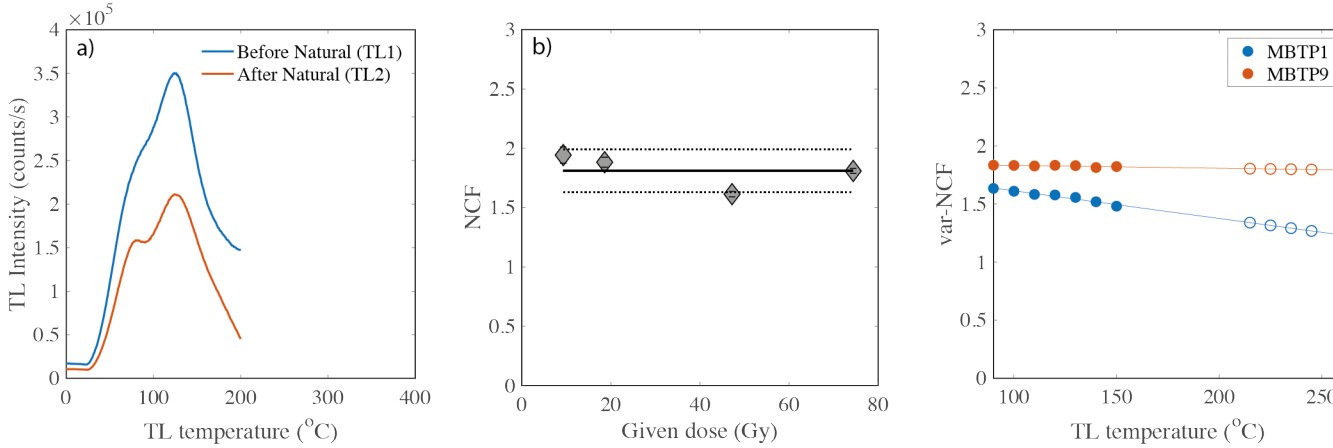

**Figure 8: a) The lower temperature TL before (TL₁) and after (TL₂) natural TL measurement (up to 450 °C) of samples MBTP1 and MBTP9. b) NCF for TL signal (integrated over 90-120 °C) for different given doses. The data are scattered and vary within ±10%, which possibly suggest the NCF is dose independent c) plot of var-NCF (=TL₁/TL₂) at different temperature (90-150 °C) along TL glow (solid circles), and extrapolated to calculate NCF in the region of interest (210-250 °C; empty circles). The values were calculated by taking the average of three measurements.**

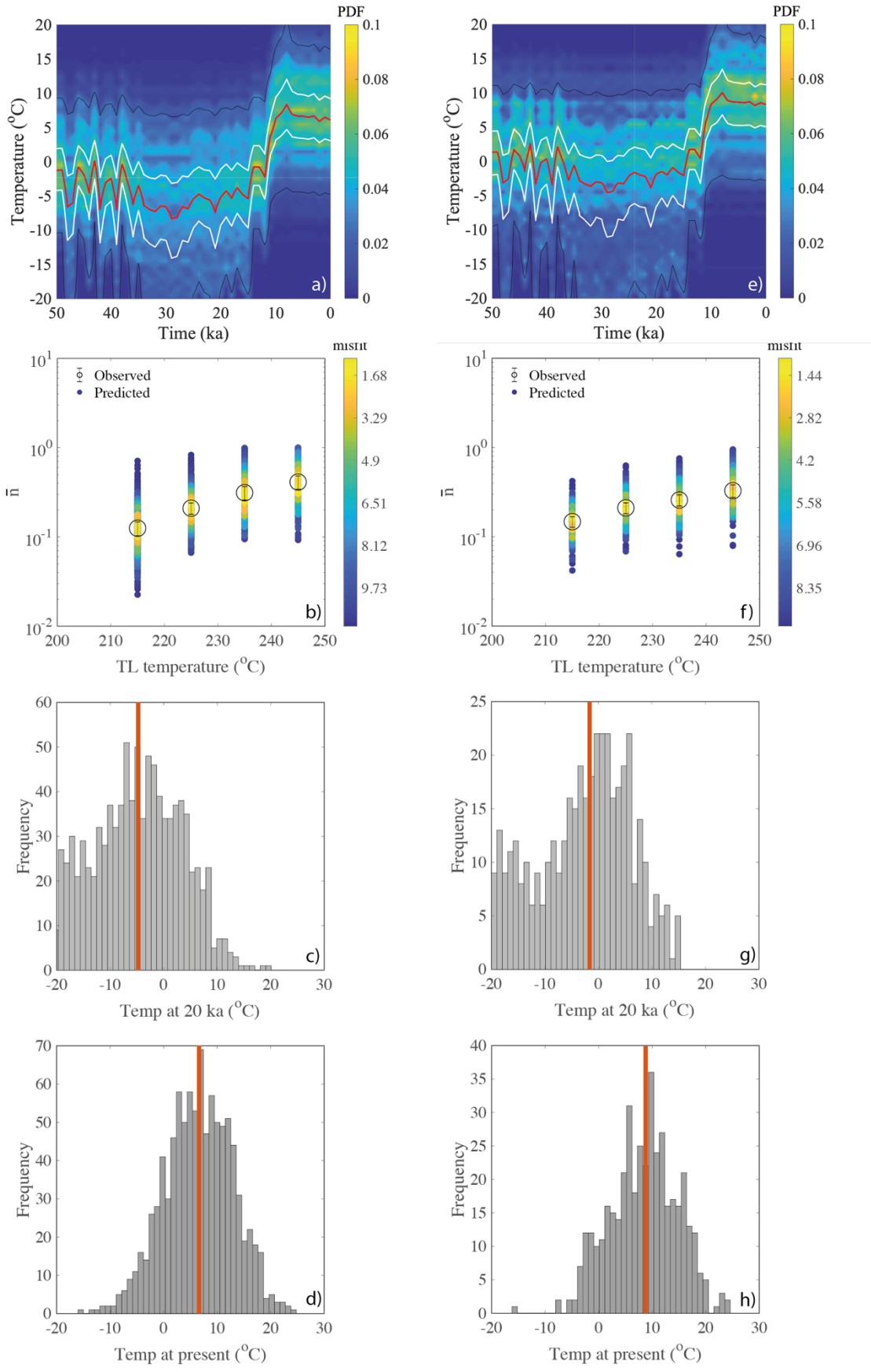

**Figure 9: Inferred rock surface temperature history for sample MBTP1 and MBTP9 collected in the Mont Blanc massif at an altitude of ~2.0-2.5 km obtained through inverse modeling of TL data as described in section 4.4. a-d are for the sample MBTP1. a) is the probability density function. The red line, white lines and black lines are the predicted median, 1σ and 2σ confidence intervals. b) is the plot of observed TL ($\bar{n}_{obs}$) and modeled TL (predicted TL through inverse modeling). c) and d) are histograms of temperature at 20 ka and present respectively. e-h are the result of same analysis for sample MBTP9.**

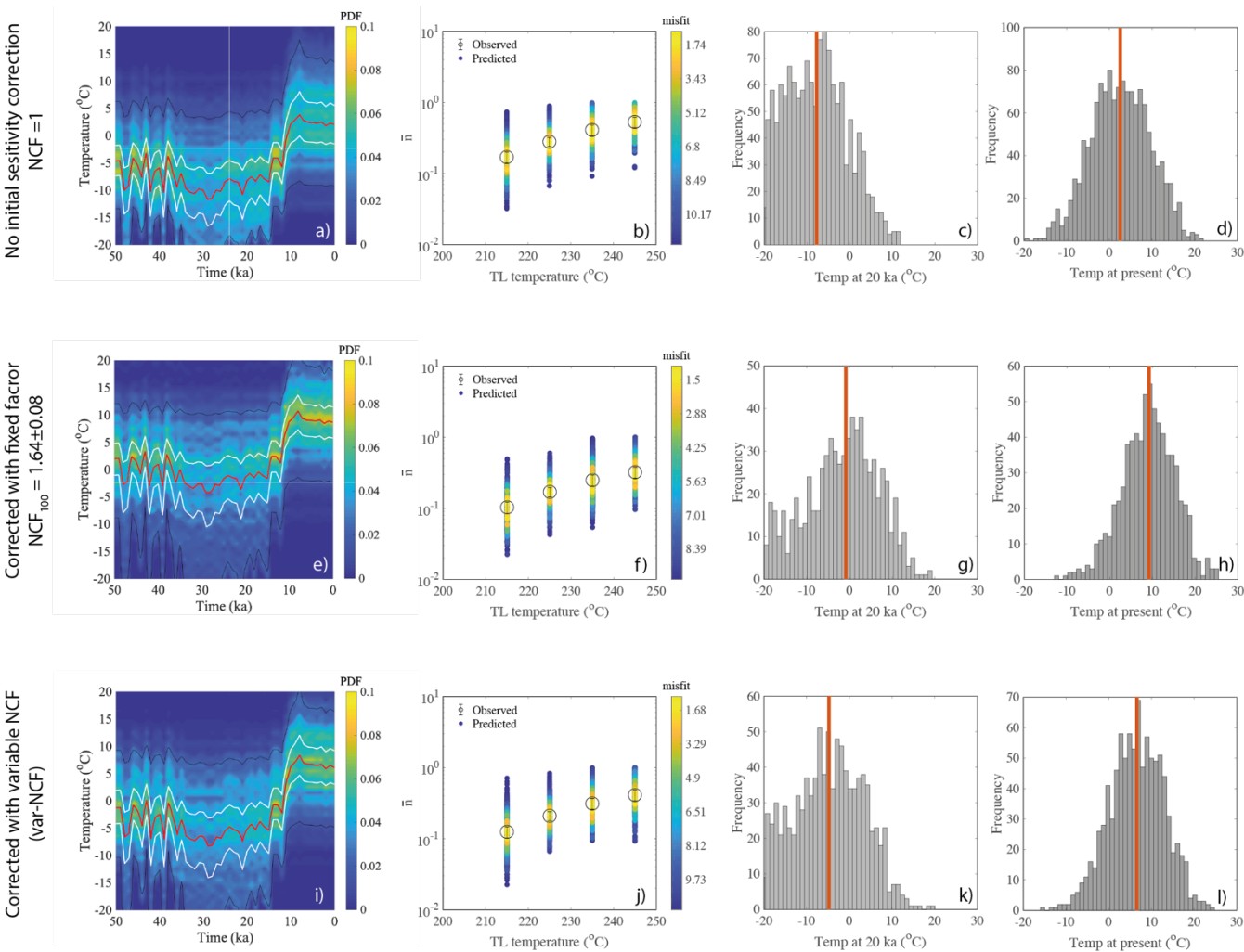

**Figure 10: The impact of initial sensitivity correction for past temperature prediction to sample MBTP1 for three different scenarios, 1) no initial sensitivity correction i.e. NCF =1, 2) initial sensitivity corrected with fixed NCF at 100 °C (=1.64±0.08) for all thermometers, and, 3) initial sensitivity corrected to all TL thermometers with var-NCF for all TL thermometers (i.e. the selected method). a, e and i are the probability distributions, and b, f and j are observed and predicted TL plots for all three scenarios respectively. c, g, k and d, h, l are the histogram of predicted temperature for all accepted paths at 20 ka and the present day for all three scenarios respectively.**

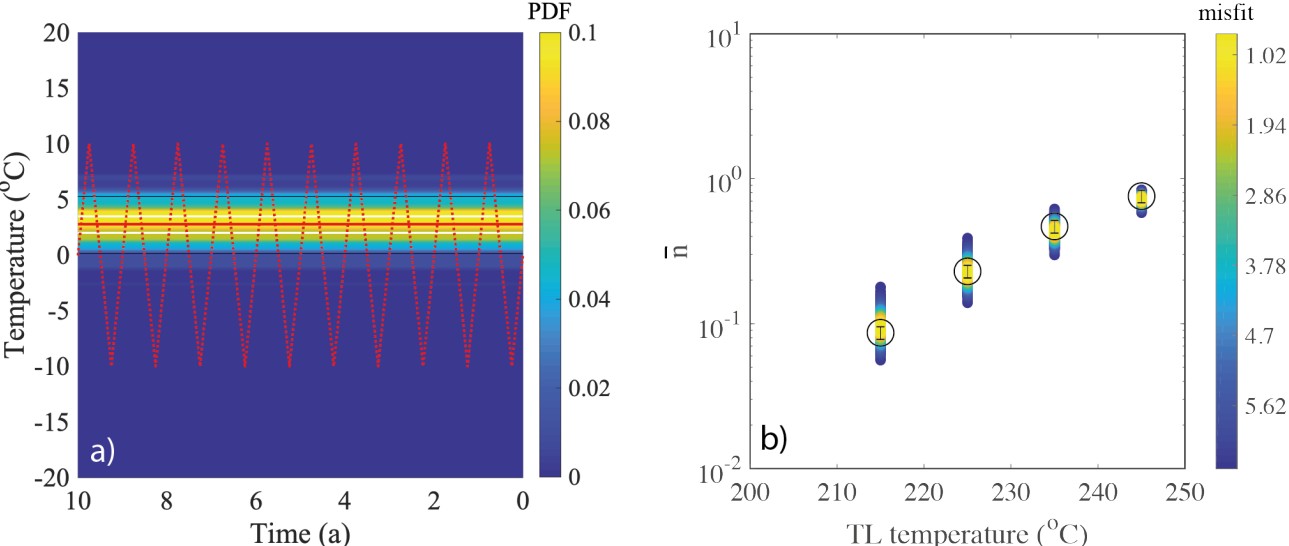

**Figure 11: Result of synthetic test for annual fluctuation of temperature. a) probability density plot. The dotted redline is the actual thermal history. Solid red line is the predicted median of the isotherms. white and black lines are the 1σ and 2σ confidence intervals. Fig. b shows the plot of observed ($\overline{n}_{obs}$) and modeled TL values (predicted TL through inverse modeling) for all TL thermometers.**

**Table1: List of kinetic parameters that describe growth (Ḋ, $D_0$, a), thermal decay (E, s, b) and athermal decay (ρ′) and natural TL (observed) trapped charge populations ($\bar{n}_{obs}$) for the four thermometers (210-250 °C, 10 °C interval) for sample MBTP1 and MBTP9. The dose rate (Ḋ) values were taken from Lehmann et al. (2020). Note that in the thermal decay parameters (*E*, *s* and *b*) no errors are mentioned. The mean values of these parameters are calculated from the distribution and an arbitrary error of 5% was considered.**

| | TL (°C) | $\dot{D}$ (Gy/ka) | Growth $D_\tau$ (Gy) | Growth $a$ | Thermal decay $E$ (eV) | Thermal decay $\log_{10}(s)$ | Thermal decay $b$ | Athermal decay $\log_{10}(\rho')$ | Natural TL ($\bar{n}_{obs}$) Uncorrected | Natural TL ($\bar{n}_{obs}$) NCF | Natural TL ($\bar{n}_{obs}$) Corrected |
|---|---|---|---|---|---|---|---|---|---|---|---|
| **MBTP1** | 210-220 | | 766 ± 51 | 1.00 ± 0.09 | 1.24 | 11.62 | 1.46 | -6.02 ± 0.08 | 0.17 ± 0.03 | 1.36 ± 0.04 | 0.13 ± 0.02 |
| | 220-230 | | 690 ± 46 | 1.00 ± 0.11 | 1.28 | 11.69 | 1.45 | -6.29 ± 0.14 | 0.28 ± 0.04 | 1.34 ± 0.03 | 0.21 ± 0.03 |
| | 230-240 | 7.39 ± 0.16 | 638 ± 43 | 1.00 ± 0.13 | 1.31 | 11.75 | 1.45 | -7.10 ± 0.94 | 0.41 ± 0.07 | 1.31 ± 0.03 | 0.31 ± 0.05 |
| | 240-250 | | 559 ± 40 | 1.00 ± 0.26 | 1.35 | 11.79 | 1.45 | <-20 ± 0 | 0.53 ± 0.09 | 1.29 ± 0.03 | 0.41 ± 0.07 |
| **MBTP9** | 210-220 | | 773 ± 41 | 1.00 ± 0.03 | 1.25 | 11.63 | 1.49 | -6.13 ± 0.09 | 0.26 ± 0.03 | 1.73 ± 0.08 | 0.15 ± 0.02 |
| | 220-230 | | 680 ± 37 | 1.00 ± 0.01 | 1.29 | 11.72 | 1.49 | -6.18 ± 0.10 | 0.36 ± 0.05 | 1.72 ± 0.08 | 0.21 ± 0.03 |
| | 230-240 | 7.07 ± 0.15 | 625 ± 40 | 1.18 ± 0.33 | 1.32 | 11.79 | 1.49 | -6.33 ± 0.17 | 0.44 ± 0.06 | 1.71 ± 0.09 | 0.26 ± 0.04 |
| | 240-250 | | 502 ± 36 | 1.10 ± 0.27 | 1.36 | 11.85 | 1.49 | -6.51 ± 0.24 | 0.56 ± 0.08 | 1.70 ± 0.09 | 0.33 ± 0.05 |