# Peer review of "Surface paleothermometry using low temperature thermoluminescence of feldspar"

_Climate of the Past, 2019_

## Referee Comment (RC1) · Anonymous Referee #1 · 24 Mar 2020

General comments:

The authors are investigating an interesting question–whether luminescence signals from K-feldspars in bedrock might archive changes in recent temperatures at Earth's surface. More specifically, they ask whether we can resolve recent changes in temperature periodicity. This question is important and worth pursuing.

I commend the authors for the layout of this study. Their approach involving sensitivity analyses, calibration to sample specific kinetics and attention to climatic complexity have resulted in an interesting manuscript with potential for significant scientific impact.

However, the present work needs significant clarification and expansion to yield a robust estimate of past temperatures. Specifically, the authors must determine how changes in amplitude, period, and mean temperature influence luminescence signal growth and depletion in a holistic way. Currently, the treatment is partial. Once this is done, the authors should give a more direct comparison of actual and predicted temperature histories in order for the reader to better examine the predictive success of the model.

These points and others are detailed below.

1/22: "Earth's climate fluctuates in a cyclic way" While there are many internal cycles to climate systems, this characterisation might be too simplistic, especially at the timescales involved here ($\sim 10^1 - 10^2$ kyr), where abrupt periods of change are common. Temperature changes during the Holocene, for example, can hardly be approximated as cyclic.

1/35: "equivalent diffusion temperature that is always higher than the actual mean temperature" This statement, while true, gives the false impression that this is an intractable bias. For a system with well-characterised diffusion kinetics, the relationship between a given temperature history (e.g., a forward model) and the EDT is well known. In other words, paleothermometry using the He-3 paleothermometry technique must rely upon comparisons against prescribed temperature histories in the same way as paleothermometry with luminescence techniques. This is not a comparative disadvantage of the noble gas technique, but a similar limitation as faced in the current study.

2/6: The distinction between thermochronology and paleothermometry is not entirely clear in the language of this study. If what you aim to resolve is the temporal variation of temperature through time, you are describing thermochronology. If instead, you mean to resolve a past temperature which is representative of some time period (the measurement of which will be affected by seasonal variability and so on), then what you are describing is paleothermometry. I would encourage more precision when you describe these concepts.

2/12-21: Please add corresponding references for these observations.

2/36ff: The physical meaning of this model is unclear. This is obviously of fundamental importance, as the kinetic model that is chosen will determine all predictions of past thermal history.

From Eq. 1, it would seem that the authors expect to model some number of individual traps, each with a singular values for D0, E, s, s-tilde, rho', a, and b. All of these traps are modelled as disconnected.

And yet, the transition in parameter values from one measurement temperature bin to another is smooth. This is true for D0, for rho', for E, for s, and for b within Fig. 1. This observation strongly suggests continuity in the underlying kinetics, not only for trap depth(s) but for the system as a whole. To fit each measurement bin as a separate and disconnected trap seems suspect. A unified treatment would be preferable.

Another issue to address is whether the same recombination centers are accessed by this distribution of traps during athermal fading. If so, as would seem unavoidable to some degree, rho should be kept constant (the density of centers being a property of the material). rho' can then be related to the underlying activation energy via the alpha term. This should be attempted for internal consistency.

According to the second term on the RHS of Eq. 1, it seems that the authors model thermally-activated recombination locally (since the term is dependent upon the nearest neighbor distribution). If so, then observations of signal loss at room temperature could also be caused by this pathway. This deserves comment.

3/29-30: Can we be confident that the kinetic parameters pertain to geologic timescales? Specifically, is there good evidence or reason to think that mixed order kinetics are predicted at low temperatures and long timescales (natural) as well as high temperatures and short timescales (lab)? Competition effects, for example, could easily produce observations of b > 1 for lab measurements whereas the concentration
of charge activated on natural timescales would be orders of magnitude smaller.

4/19 and Fig. 3: I find this figure a little difficult to interpret. In particular, the way that you have defined 'memory time' should be a bit clearer. If I've understood the meaning of this metric correctly, one suggestion would be to compare everything against 't-change = 100ka' or to extend the x-axis to include 't-change=150ka.' By doing this, you could then visually show the meaning of the 't-memory' by comparing two horizontal lines, one at t-change=150ka (or 100ka, depending), and the other at the asterisk height. You could then annotate this difference as 20%.

It would be good to consider also the influence of measurement uncertainty. Accurately resolving the difference between [n-bar] = 2.0e-3 and 2.4e-3 would likely involve a lot more relative uncertainty than discerning between, say, [n-bar] = 0.5 and 0.6.

4/30-31: '10 - 100 kyr timescales' This should be a bit more specific. During the Quaternary, the 100kyr and 41kyr periods were most dominant (e.g., Raymo et al., 2006; Hinnov, 2013).

Fig. 4: This is a nice figure. Please adjust so that the 'thermometer' labels correspond to each panel. For example, in panels j-l, if the reader uses the labels on the far right-hand side, they might conclude that the bottom-most series in panel k refers to the 220-230C chronometer.

5/7: 'implies a gradient' I would say this behavior more accurately 'results' from a gradient in thermal stability, given that the kinetic parameters are imposed and known.

5/9: 'the thermal stability decreases with increasing temperature.' I disagree with this statement. The thermal stability for a given TL thermometer is known and fixed in your setup. So it is not the stability that is changing, but rather the probability of detrapping.

5/10: 'the higher temperature TL thermometers remain relatively insensitive to such periodic temperature forcing.' This is misleading. The insensitivity of the higher-T thermometers reflects the mean temperature values that you have chosen. If the temperature oscillated about a higher mean value, the same periodic filling/emptying behavior would be seen with the high-T thermometers. This is evident in Fig. 4 panels j-l, where different thermometers oscillate with comparable magnitude when a range of mean temperatures are tested.

5/13: '10 ka to 1 Ba' If Ba represents 'billion years,' please change to 'Ga.'

5/15: 'For P « tau' This comparison must be qualified. The lifetime (tau) will depend on a chosen, singular temperature value. Choosing a singular temperature value for oscillating temperature requires some simplification that is not described (e.g., mean temperature? EDT?). Please clarify this issue.

5/16: 'This result implies that smaller periods (<1 ka) do not influence trapped charge equilibrium levels in an oscillating fashion and cannot be differentiated from the trapped charge population resulting from an isothermal condition.'

This statement is incomplete and only conditionally true. For argument's sake, assume that the 200C TL thermometer has a lifetime of 10 ka at 0C. If the ambient temperature oscillated with an arbitrarily large amplitude (say 100C to make the point obvious) but with a period of only 1 ka or 100 yr, you would find that the fractional saturation would oscillate in response to the temperature forcing, depleting completely and then partially regenerating.

This won't happen if the temperature period is much smaller than the growth timescale (D0/Ddot). If that is the case, then the thermal imprint upon the sample approaches a steady value determined by the maximum temperature experienced.

Temperature amplitude and the relationship between the forcing period and sample growth timescale both matter. To make this comparison between lifetime and period, these factors must be incorporated.

5/19: 'remains correlated' and 'deviates from...temperature forcing' The meanings of these statements are unclear. n-bar behavior is distinct between isothermal and oscillating temperature histories for all periods; it is not an issue of matching and not matching, except for the highest-temperature systems, which are insensitive to the temperatures prescribed here. Please be more specific with these observations and, following from the previous comment, please do incorporate growth timescales, as these are of obvious relevance here.

5/24: 'Therefore, temperature variations can be reconstructed...' Just to reiterate, you must demonstrate the complex relationship among mean temperature, temperature amplitude, trap stability and regenerative timescales before attempting to reconstruct temperature variability. Additionally, what has been shown in Fig. 4 is that, for a given amplitude, different periods leave different imprints upon the shown thermometers. You have not yet demonstrated that you can accurately reconstruct differences in variability. Moreover, the results from Fig. 5 (panels b, c) seem to indicate that you cannot easily differentiate between various amplitudes or periods.

5/31-33: 'This ensures that complex thermal histories...can be reconstructed.' Following from the previous comment, this is not yet demonstrated.

6/11: 'considering that the other parameters are identical.' Unclear what this means. From Fig. 1, it would seem that the kinetic parameters other than the thermal parameters ($E$, $s$) vary between the thermometers. Or do you mean something else?

Fig. 7: This is not the most informative way to show your predictive ability. Unlike with thermochronology studies, where the T-t path is really the predicted feature, what you are more accurately doing is predicting the temperature minimum and amplitude (e.g., ll. 5/24-29). So, it would be much more informative to see these values, actual and predicted.

Additionally, the current figure makes it appear as if you are able to resolve the fine structure of the T-t series, which of course you are not.

7/16-18: '[Fig. 7a,b,c shows] it is possible to recover all three thermal histories within

the 1-sigma confidence level.' My previous comment will apply here as well. What matters for this experiment is the degree to which you are able to predict amplitude and minimum.

To demonstrate that you could recover an arbitrary thermal history well, you would need a different test.

8/7: 'This restrict...' Grammar

8/15: This dose range, with upper doses at 0.9 and 1.9 kGy may not be sufficient to observe saturation in K-feldspar TL. Please demonstrate that these signals are saturating or add greater doses to better constrain lab saturation intensity.

8/33: 'no effect for doses below 100 Gy' Wouldn't the far more important question be whether there is sensitivity change above 100 Gy? After all, the majority of given doses are above 100 Gy and these responses allow you to determine the saturation values.

9/13-15: 'The rationale here is that all temperatures follow the delta O-18 data, but the amplitude...and mean temperature are unknown.' Please make clear that this is a stated assumption and not an inference from studies (unless it is, in which case state that). I think is not obvious that the climate signal on Mont Blanc would mirror a Greenland ice core signal, so this assumption probably warrants justification.

Fig. 9: Is Fig. 9 referenced in the main text?

Fig. 9 and 10: As with Fig. 7, please recast to compare the predicted and actual values for the amplitude and base temperatures as these are the variables being investigated.

9/28: 'can constrain thermal history of ∼50kyr.' I do not think this has been demonstrated yet, as an extension of my comments regarding pg. 5.

Also, unclear what 'A higher temperature fluctuation' means and whether you've actually shown this.

10/14: 'At those depths [of 7 and 62 cm], mean temperature should be constant.' Certainly, there will be seasonal temperature variability at a depth of 7 cm. Is this what you meant to say?

10/17: For inverse modeling of a natural sample, the time-temperature histories were completely random' If I have understood the previous text correctly, the histories are very much not random, but are tied to the Greenland delta O-18 temperature proxy with variability only in amplitude and initial temperature. Please reword.

Discussions generally: I won't comment much on these inferences about past climate systems at this stage, because I think it will be very important to first demonstrate model success in capturing simple variations within a periodic forcing model, which, at this stage, has not been done.

———————————————————

---

## Referee Comment (RC2) · Anonymous Referee #2 · 9 Apr 2020

This manuscript proposes a new method to study/reconstruct paleo-temperature using TL of feldspar from rock surface. Reconstructing paleo-temperature is an important topic in climatic change study, so the attempt of this study is important and worth for publication. However, there are some technical issues that are not settled well, which prevents convincing me that this is achievable. Here I summarise my major concerns.

1) Kinetic model: the authors considered three processes in their model, including dosing (growth), thermal decay and athermal decay (fading), which are represented by different terms in their equation 1. They then estimated the parameters based on TL measurements in different ways. For both the growth curve parameters (D0 and a) and fading parameters (rho'), my understanding is that they were based on the signals from different integrals of the TL glow curve (e.g., 200 – 250 C in 10C interval). However, for the thermal decay parameters (E, s and b), they used a $T_m$-$T_{stop}$ method, in which the signal from each temperature interval is obtained from subtracting consecutive fractional glow curve. That means, the signals they used to estimate present-day charge population ($\bar{n}$), growth curve and fading are based on simple integral of TL signals at different intervals, which obviously are a mix of signals from a range of trapping energy levels, but the thermal kinetic parameters are based on single (or narrow-range) trapping energy levels. That says, the authors did not separate the TL signals for constructing their model using a similar way (Tm-Tstop method) that they did for estimating the thermal decay parameters. This is problematic, as the combination of different trapping levels are not linear, so their model (equation 1) simply becomes invalid when the signals being analysed are associated with a range of trapping levels are analysed. One need to makes sure that the different parameters in Equation 1 are all obtained from the same signal associated to a single or narrow-range energy level. I am not use if this can be achieved as the combination of MAR protocol (and fading test) and Tm-Tstop would be very difficult to achieve.

2) The authors simply assuming feldspar consist a continuous trapping energy level. However, it has been commonly accepted that band-tail states play important role in the luminescence process (including TL and IRSL) in feldspar (Poolton et al., 2002). I do not see any reason to discard the band-tail states from their model.

3) How sensitive is the model to dose rate ($\dot{D}$)? The dose rate would play an important role in filling the traps during natural process, so I would expect that it will somehow influence the model results. Unfortunately, the dosimetry of the samples is poorly described. How did the author estimate the dose rate of K-feldspar? The author appears to simply crush the rock and select 150 – 250 um grain size range. What are the original grain sizes of the K-feldspar minerals in the rock? Did the authors make any rock slide to investigate this? This is critical as there are a large contribution of internal dose rates for K-feldspar.

4) The authors applied the NCF method to overcome sensitivity change issue for TL measurements. They do realise the limitation of this method as it is based on extrapolation of the NCF values from low temperature to high temperature

region, which is unreliable. Although the authors tested the effect of initial sensitivity changes on the modelling results and found very little changes for their sample, it does not guarantee that this applies to other samples and situations. The reason that the sensitivity changes did not affect the results is simply because that their samples are young and the growth of signal still lies on the linear part of growth curve, so any systematic changes in the sensitivity result in a proportional changes of different signal integrals. For older samples or high-De samples, however, this may result in non-proportional changes among different signal integrals (because the different D0 values for different integrals), and, hence, different model results in paleo-temperature. This potential problems should be appropriately acknowledged as at present it gives false impression that the initial sensitivity change does not matter.

Minor comments:

5) Line 21: Credits should be given to Li and Li (Li and Li, 2012) who firstly proposed the idea of multiple-thermometers using TL (although not implemented in their study), and they also first introduced the rate equation to investigate the effect of single growth and saturation on OSL-thermochronology.

6) Figure 1: The authors should at least provide some typical TL glow curves for their samples before showing the kinetic results.

7) Figure 1c: The fading parameter (rho') shows systematic change as a function of temperature up to 280C, but it suddenly become 'no fading'. This is surprising. The different integral signals represent a continuous mixture of signals of different athermal features, why one can obtain a sudden change in the fading? Is it because that the fading rate has large uncertainty range consistent with zero fading? In this case, it would be problematic to say 'no fading', as statistically it is also like to be 'fading'.

8) Figure 8b: Why not plot the results for other temperature range (e.g., 120 – 150 C)?

9) Figure 8c: what are the errors for the NCF at high temperature range (200 – 250C)? Have you incorporated the NFC errors into the final results?

10) Table 1: Why there are no errors for E, s and b? Why an arbitrary error of 5% is assumed, rather than their actual analytical errors?

11) Figure S3: why there is only one natural point but 3 regenerative points for each thermometer? Did the author just measure one aliquot for natural?

References

Li, B., Li, S.-H., 2012. Determining the cooling age using luminescence-thermochronology. Tectonophysics 580, 242-248.

Poolton, N.R.J., Ozanyan, K.B., Wallinga, J., Murray, A.S., Bøtter-Jensen, L., 2002. Electrons in feldspar II: a consideration of the influence of conduction band-tail states on luminescence processes. Physics and Chemistry of Minerals 29, 217-225.

---

## Author Comment (AC1) · 3 May 2020

General comments:

The authors are investigating an interesting question–whether luminescence signals from K-feldspars in bedrock might archive changes in recent temperatures at Earth's surface. More specifically, they ask whether we can resolve recent changes in temperature periodicity. This question is important and worth pursuing.

I commend the authors for the layout of this study. Their approach involving sensitivity analyses, calibration to sample specific kinetics and attention to climatic complexity have resulted in an interesting manuscript with potential for significant scientific impact.

We appreciate the reviewer for discerning the potential of this work.

However, the present work needs significant clarification and expansion to yield a robust estimate of past temperatures. Specifically, the authors must determine how changes in amplitude, period, and mean temperature influence luminescence signal growth and depletion in a holistic way. Currently, the treatment is partial. Once this is done, the authors should give a more direct comparison of actual and predicted temperature histories in order for the reader to better examine the predictive success of the model.

Please see the replies below.

These points and others are detailed below.

1/22: "Earth's climate fluctuates in a cyclic way" While there are many internal cycles to climate systems, this characterisation might be too simplistic, especially at the timescales involved here (~10^1 - 10^2 kyr), where abrupt periods of change are common. Temperature changes during the Holocene, for example, can hardly be approximated as cyclic.

We have changed the sentence to… "*Earth's climate fluctuates, from seasonal to million-year time scales driven by Earth's orbital processes and rare shifts and extreme climate transitions over timescales of $10^3$ to $10^5$ years*".

1/35: "equivalent diffusion temperature that is always higher than the actual mean temperature" This statement, while true, gives the false impression that this is an intractable bias. For a system with well-characterised diffusion kinetics, the relationship between a given temperature history (e.g., a forward model) and the EDT is well known. In other words, paleothermometry using the He-3 paleothermometry technique must rely upon comparisons against prescribed temperature histories in

the same way as paleothermometry with luminescence techniques. This is not a comparative disadvantage of the noble gas technique, but a similar limitation as faced in the current study.

This is true. We meant that a single thermometric system (OSL or $^3$He) will always provide a single equivalent temperature (like EDT in $^3$He) for a complex thermal history whereas a multi thermometric system (here TL) with different temperature and time sensitivities will have the potential to infer a more complex thermal history. The following text has been added: "*For a system with well-characterised diffusion kinetics, the relationship between a given temperature history and the EDT is well known, so the paleotemperature can be corrected and estimated.*"

2/6: The distinction between thermochronology and paleothermometry is not entirely clear in the language of this study. If what you aim to resolve is the temporal variation of temperature through time, you are describing thermochronology. If instead, you mean to resolve a past temperature which is representative of some time period (the measurement of which will be affected by seasonal variability and so on), then what you are describing is paleothermometry. I would encourage more precision when you describe these concepts.

Paleothermometry is a methodology for determining past temperature, and thermochronology is the study of the thermal evolution of rocks, and they both rely on the same principle. However, thermochronology is more commonly used for rock cooling related to exhumation. So to avoid confusion, we now use paleothermometry throughout the manuscript. We only refer to thermochronometry when we refer to previous work, which was developed for thermal evolution of rocks associated with exhumation.

2/12-21: Please add corresponding references for these observations.

Amended.

2/36ff: The physical meaning of this model is unclear. This is obviously of fundamental importance, as the kinetic model that is chosen will determine all predictions of past thermal history.

From Eq. 1, it would seem that the authors expect to model some number of individual traps, each with a singular values for D0, E, s, s-tilde, rho', a, and b. All of these traps are modelled as disconnected.

And yet, the transition in parameter values from one measurement temperature bin to another is smooth. This is true for D0, for rho', for E, for s, and for b within Fig. 1. This observation strongly suggests continuity in the underlying kinetics, not only for trap depth(s) but for the system as a whole. To fit each measurement bin as a separate and disconnected trap seems suspect. A unified treatment would be preferable.

Several studies suggest that broad TL glow curve from feldspar arises from a continuous distribution of trapping energies, which is suggested by several methods, like $T_m$-$T_{stop}$, the initial rise method, and analysis of fractional glow curves (Biswas et al., 2018; Grün and Packman, 1994; Pagonis et al., 2014; Strickertsson, 1985). Regardless, it is difficult to isolate a single trap with distinct kinetic parameters. Instead we assign the most probable kinetic parameters for each thermometer (glow curve temperature)

along the TL glow curve and in this manner we determine kinetic values in a continuous, rather than in a disconnected manner. This is the method that we have adopted here and in Biswas et al. (2018). We then arbitrarily choose 10 °C TL temperature windows as distinct thermometers. A continuous distribution of trapping energies can be assumed as the sum of a large number of discrete traps (Pagonis et al. 2014). Thus a continuous distribution of trapping energies is discretized as shown in the figure below.

[Figure]

**Fig. X: The evaluated continuous distribution of trap depth (E) of sample MBTP1 (circles with error bar) and its discretization in 10 °C windows (green box; width is 10 °C and height 5 % of the median value).**

Another issue to address is whether the same recombination centers are accessed by this distribution of traps during athermal fading. If so, as would seem unavoidable to some degree, rho should be kept constant (the density of centers being a property of the material). rho' can then be related to the underlying activation energy via the alpha term. This should be attempted for internal consistency.

Here we only use $\rho'$, which includes alpha as $\rho' = \frac{4\pi\rho}{3\alpha^3}$. Alpha controls the rate of fading through the lifetime $\tau = s^{-1}\exp(\alpha r)$. It is expected that with increasing activation energy fading rate ($\rho'$) should decreases as the tunnelling depth increases. Indeed we get similar relationship between fading rate ($\rho'$) and activation energy (Table 1) as shown in figure below.

[Figure]

**Fig. XX: Plot of distance dependent fading rate ($\rho'$) and activation energy ($E$) of five thermometers (200-250 °C, 10 °C interval) of sample MBTP9.**

According to the second term on the RHS of Eq. 1, it seems that the authors model thermally-activated recombination locally (since the term is dependent upon the nearest neighbor distribution). If so, then observations of signal loss at room temperature could also be caused by this pathway. This deserves comment.

Thermal loss of the TL signals which is a more diffusive process (athermal) can occur through the conduction band. However, the power term $b$, accounts for the nonlinearity (delayed) arise due to presence of band-tail states. However, athermal fading is known as a tunnelling process. Separate treatments of thermal and athermal loss has adopted successfully in several previous studies (Guralnik et al. 2015 for IRSL of feldspar; Biswas et al. 2018 for TL of feldspar).

3/29-30: Can we be confident that the kinetic parameters pertain to geologic timescales? Specifically, is there good evidence or reason to think that mixed order kinetics are predicted at low temperatures and long timescales (natural) as well as high temperatures and short timescales (lab)? Competition effects, for example, could easily produce observations of b > 1 for lab measurements whereas the concentration of charge activated on natural timescales would be orders of magnitude smaller.

This is a difficult question and can never be answered in true sense but the premises of the approach used here was validated in Biswas et al. (2018), by successful recovering temperatures experienced by rocks in the KTB borehole, which have been in thermal steady state for several millions of years. We have included that statement in the manuscript.

4/19 and Fig. 3: I find this figure a little difficult to interpret. In particular, the way that you have defined 'memory time' should be a bit clearer. If I've understood the meaning of this metric correctly, one suggestion would be to compare everything against 'tchange = 100ka' or to extend the x-axis to include 't-change=150ka.' By doing this, you could then visually show the meaning of the 't-memory' by comparing two horizontal lines, one at t-change=150ka (or 100ka, depending), and the other at the asterisk height. You could then annotate this difference as 20%.

*Amended as recommended. In the figure the 20% changes have been demonstrated clearly and text has been added in the figure title (Fig. 3).*

It would be good to consider also the influence of measurement uncertainty. Accurately resolving the difference between [n-bar] = 2.0e-3 and 2.4e-3 would likely involve a lot more relative uncertainty than discerning between, say, [n-bar] = 0.5 and 0.6.

*Typical photon counts of the present sample for near saturation ($\bar{n} = 1$) is >$10^5$ counts per second (cps), our instrumentation can resolve a few hundreds of cps with 20% uncertainty ($\bar{n}$ of the order of $10^{-3}$). This is discussed in Section 4.3 as "Typically, the minimum detectable limit for the present instrument is ~ 300 photon counts per second (cps) considering the signal should be three times of background level which is ~ 100 cps. The present highly luminescent feldspar has maximum photon count of ~$10^6$ cps. This restrict to use the TL signals up to ~$10^{-3}$ % of maximum TL signals"*

4/30-31: '10 - 100 kyr timescales' This should be a bit more specific. During the Quaternary, the 100kyr and 41kyr periods were most dominant (e.g., Raymo et al., 2006; Hinnov, 2013).

*We have changed the sentence to "In nature, climate varies on a daily and seasonal basis and follows periodic variations in the Earth's orbit, known as Milankovitch cycles, at 25.77, 41 and 100 kyr"*

Fig. 4: This is a nice figure. Please adjust so that the 'thermometer' labels correspond to each panel. For example, in panels j-l, if the reader uses the labels on the far righthand side, they might conclude that the bottom-most series in panel k refers to the 220-230C chronometer.

*Amended as recommended (Fig. 4).*

5/7: 'implies a gradient' I would say this behavior more accurately 'results' from a gradient in thermal stability, given that the kinetic parameters are imposed and known.

*Amended.*

5/9: 'the thermal stability decreases with increasing temperature.' I disagree with this statement. The thermal stability for a given TL thermometer is known and fixed in your setup. So it is not the stability that is changing, but rather the probability of detrapping.

*Thanks for pointing this out. The sentence has been reworded to "This is because the probability of detrapping increases with increasing temperature".*

5/10: 'the higher temperature TL thermometers remain relatively insensitive to such periodic temperature forcing.' This is misleading. The insensitivity of the higher-T thermometers reflects the mean temperature values that you have chosen. If the temperature oscillated about a higher mean value, the same periodic filling/emptying behavior would be seen with the high-T thermometers. This is evident in Fig. 4 panels j-l, where different thermometers oscillate with comparable magnitude when a range of mean temperatures are tested.

*The sentence has been changed to "Finally, the higher temperature TL thermometers (near to 300 °C) remain relatively insensitive to such periodic temperature forcing ($T_{mean}$ up to 30 °C); with increasing $T_{mean}$ the higher temperature TL thermometers become more responsive."*

5/13: '10 ka to 1 Ba' If Ba represents 'billion years,' please change to 'Ga.'

*Amended.*

5/15: 'For P « tau' This comparison must be qualified. The lifetime (tau) will depend on a chosen, singular temperature value. Choosing a singular temperature value for oscillating temperature requires some simplification that is not described (e.g., mean temperature? EDT?). Please clarify this issue.

*The sentence has been clarified to "For $P \ll \tau$ (e.g. Fig. 4d where P=1 ka and $\tau$ spans ~10 ka to 1 Ga for the 200 to 300 °C TL thermometers for $T_{mean}$= 0 °C), the value of $\bar{n}_{osc}$ exhibits small fluctuations but always remains lower than $\bar{n}_{iso}$"*

5/16: 'This result implies that smaller periods (<1 ka) do not influence trapped charge equilibrium levels in an oscillating fashion and cannot be differentiated from the trapped charge population resulting from an isothermal condition.'

This statement is incomplete and only conditionally true. For argument's sake, assume that the 200C TL thermometer has a lifetime of 10 ka at 0C. If the ambient temperature oscillated with an arbitrarily large amplitude (say 100C to make the point obvious) but with a period of only 1 ka or 100 yr, you would find that the fractional saturation would oscillate in response to the temperature forcing, depleting completely and then partially regenerating.

This won't happen if the temperature period is much smaller than the growth timescale (D0/Ddot). If that is the case, then the thermal imprint upon the sample approaches a steady value determined by the maximum temperature experienced.

Temperature amplitude and the relationship between the forcing period and sample growth timescale both matter. To make this comparison between lifetime and period, these factors must be incorporated.

*The sentence has been modified to "This result implies that smaller periods (<1 ka) and $T_{mean}$ (<30 °C) do not influence trapped charge equilibrium levels of 200 to 300 °C TL thermometers in an oscillating fashion and cannot be differentiated from the trapped charge population resulting from an isothermal condition. We must mention here, if the amplitude of oscillation increases the oscillating response to trapped charge equilibrium levels will be relatively prominent."*

5/19: 'remains correlated' and 'deviates from...temperature forcing' The meanings of these statements are unclear. n-bar behavior is distinct between isothermal and oscillating temperature histories for all periods; it is not an issue of matching and not matching, except for the highest-temperature systems, which are insensitive to the temperatures prescribed here. Please be more specific with these observations and, following from the previous comment, please do incorporate growth timescales, as these are of obvious relevance here.

Clarifications have been made. The sentence has been changed to "*Similarly, the present day $\bar{n}_{osc}$ remains indistinguishable from $\bar{n}_{iso}$ when $P>>\tau$ (e.g. see the behaviour of the low temperature TL thermometer shown in Fig. 4l)*"

5/24: 'Therefore, temperature variations can be reconstructed...' Just to reiterate, you must demonstrate the complex relationship among mean temperature, temperature amplitude, trap stability and regenerative timescales before attempting to reconstruct temperature variability. Additionally, what has been shown in Fig. 4 is that, for a given amplitude, different periods leave different imprints upon the shown thermometers. You have not yet demonstrated that you can accurately reconstruct differences in variability. Moreover, the results from Fig. 5 (panels b, c) seem to indicate that you cannot easily differentiate between various amplitudes or periods.

This is discussed in section 2.2.2 and Fig. 4 where we show that the present day trapped charge population ($\bar{n}_{present}$) is highly sensitive to the mean temperature ($T_{mean}$) and less sensitive to the amplitude of oscillation ($T_{amp}$) and period ($P$). Moreover, we also mention that "*Although the $\bar{n}_{present}$ is less sensitive to the amplitude and the period, the pattern of $\bar{n}_{present}$ for different thermometers is distinguishable*". Additionally, now we have added a new synthetic test to emphasis the sensitivity of the three periodic parameter ($T_{mean}$, $T_{amp}$ and $P$) on the present day trapped charge population ($\bar{n}_{present}$) in section 3 as follows:

"We choose three arbitrary periodic thermal histories, Path1 ($T_{mean}$= 10 °C, $T_{amp}$= 10 °C and $P$= 25.77 ka), Path2 ($T_{mean}$= 20 °C, $T_{amp}$= 10 °C and $P$= 25.77 ka) and Path3 ($T_{mean}$= 10 °C, $T_{amp}$= 20 °C and $P$= 25.77 ka). For each thermal history, the present day trapped charge concentrations ($\bar{n}_{obs}$) are calculated for four TL thermometers (210-250 °C, 10 °C interval) as described in section 3.1. We then invert the TL data ($\bar{n}_{obs}$) into a thermal history as described in section 3.2. For the inverse modelling, we first generate a large number of random periodic histories (300,000) with $T_{mean}$ and $T_{amp}$ randomly varying from 0 to 50 °C, $P$ randomly varies between three cycles, 25.77, 41 and 100 ka. The reason we do not vary $P$ in a completely random fashion is that $\bar{n}_{obs}$ is less sensitive to $P$; it is difficult to resolve close periods (as discussed in section 2.2.2 and Fig. 5). The results are shown in the figure below (Fig. XXXa,b,c). Although this approach predicts the very recent temperature well (up to max 5 ka) it loses the periodic information (25.77 ka) because of the significant number of accepted thermal histories with different periods (41 and 100 ka). The same exercise was repeated but fixing the period to 25.77 ka for the inversion. The results are shown in Fig. XXXg,h,i. Interestingly, this approach enables to recover the actual solution within $1\sigma$ uncertainty. This shows that a periodic thermal history can be predicted

well if the period is known a priori; it enables to constrains $T_{mean}$ and $T_{amp}$ satisfactorily. To circumvent the limitation (period) of this method we use the $\delta^{18}O$ data to impose the shape of the thermal histories as a priori information. This is typically done for inversion problem when appropriate. We then constrain $T_{mean}$ and $T_{amp}$ of the spectrum (as discussed in section 3).

[Figure]

**Fig. XXX: Result of synthetic experiments for the three periodic thermal histories as described above. a), b) and c) are the inferred probability density functions when $T_{mean}$, $T_{amp}$ and $P$ are randomly varied. d), e), and f) depict the fit between the observed TL (obtained through forward modeling). The solid red lines show the predicted median, white lines and black lines show the 1$\sigma$ and 2$\sigma$ confidence intervals in the probability density distribution g), h) and i) are the inferred probability density functions when $T_{mean}$, and $T_{amp}$ are randomly varied but $P$ is fixed. j), k), and l) depict the fit between the observed TL (obtained through forward modeling)**

5/31-33: 'This ensures that complex thermal histories...can be reconstructed.' Following from the previous comment, this is not yet demonstrated.

See the answer above.

6/11: 'considering that the other parameters are identical.' Unclear what this means. From Fig. 1, it would seem that the kinetic parameters other than the thermal parameters (E, s) vary between the thermometers. Or do you mean something else?

We agree this was confusing. We have now removed these unnecessary words.

Fig. 7: This is not the most informative way to show your predictive ability. Unlike with thermochronology studies, where the T-t path is really the predicted feature, what you are more accurately doing is predicting the temperature minimum and amplitude (e.g., ll. 5/24-29). So, it would be much more informative to see these values, actual and predicted.

It is true that the only difference with the inversion of thermochronometric data is that we impose the shape of $\delta^{18}$O. To address the reviewer's comment we have added histograms of the two main scaling parameters: the minimum temperature $T_{base}$ (temperature at 20 ka) and the present temperature, $T_{present}$ (which is $T_{base}+T_{amp}$ as shown in supplement Eq. S9).

Additionally, the current figure makes it appear as if you are able to resolve the fine structure of the T-t series, which of course you are not.

Indeed, we cannot. Rather this figure shows the scaling of the $\delta^{18}$O curve. The reply to the previous comment should have addressed this comment.

7/16-18: '[Fig. 7a,b,c shows] it is possible to recover all three thermal histories within the 1-sigma confidence level.' My previous comment will apply here as well. What matters for this experiment is the degree to which you are able to predict amplitude and minimum.

To demonstrate that you could recover an arbitrary thermal history well, you would need a different test.

Please see the reply of previous comment. We clearly mention we need prior information for the pattern of the thermal history ($\delta^{18}$O), as typically done for inversions, but that we can constrain the temperature minimum and amplitude.

8/7: 'This restrict...' Grammar

Amended.

8/15: This dose range, with upper doses at 0.9 and 1.9 kGy may not be sufficient to observe saturation in K-feldspar TL. Please demonstrate that these signals are saturating or add greater doses to better constrain lab saturation intensity.

1.9 kGy is sufficiently high as it is greater than $2D_0$; maximum $D_0$ is less than 800 Gy (see Table 1).

8/33: 'no effect for doses below 100 Gy' Wouldn't the far more important question be whether there is sensitivity change above 100 Gy? After all, the majority of given doses are above 100 Gy and these responses allow you to determine the saturation values.

The sensitivity change in the NCF method permits determination of the equivalent dose correctly by correcting for possible sensitivity changes during a sample read out and is generally tested on low doses (see Singhvi et al. 2011).

9/13-15: 'The rationale here is that all temperatures follow the delta O-18 data, but the amplitude...and mean temperature are unknown.' Please make clear that this is a stated assumption and not an inference from studies (unless it is, in which case state that). I think is not obvious that the climate signal on Mont Blanc would mirror a Greenland ice core signal, so this assumption probably warrants justification.

*To clarify, the text has been modified as follows "It is assumed that the atmospheric temperatures of the Mont Blanc massif followed the trend observed for the Greenland ice core data over the last 60 kyr. Note that temperature increase during the last glacial cycle was synchronous with the temperature anomalies observed in Greenland (e.g., Schwander et al., 2000; van Raden et al. 2013).The rationale here is that all temperatures in the Mont Blanc massif follow the Greenland ice core $\delta^{18}O$ data but the amplitude of temperature oscillation (minimum temperature at ~20 ka to maximum temperature at the present day) and mean temperature are unknown."*

Fig. 9: Is Fig. 9 referenced in the main text?

*Now it is mentioned.*

Fig. 9 and 10: As with Fig. 7, please recast to compare the predicted and actual values for the amplitude and base temperatures as these are the variables being investigated.

*We have now added subplots (histograms) of the two parameters, temperature minimum which we define as $T_{base}$ (temperature at 20 ka) and present temperature, $T_{present}$ (which is $T_{base}+T_{amp}$ as shown in supplement Eq. S9).*

9/28: 'can constrain thermal history of ~50kyr.' I do not think this has been demonstrated yet, as an extension of my comments regarding pg. 5.

*In Fig. 3 we show that $t_{memory}$ can be up to 50 kyr for typical temperature of 10-20 °C. If the amplitude of temperature change is higher, the method will be sensitive, thereby improving our ability to constrain the thermal history constraining of thermal history will be more precise.*

Also, unclear what 'A higher temperature fluctuation' means and whether you've actually shown this.

*This is shown in Fig. 3.*

10/14: 'At those depths [of 7 and 62 cm], mean temperature should be constant.' Certainly, there will be seasonal temperature variability at a depth of 7 cm. Is this what you meant to say?

*We mean that at depths of 7 and 62 cm the rock temperature will be equilibrium with the atmospheric temperature. The sentence has been modified to "At those depths, mean temperature should be in equilibrium with atmospheric temperature (e.g. Hasler et al. 2011)"*

10/17: For inverse modeling of a natural sample, the time-temperature histories were completely random' If I have understood the previous text correctly, the histories are very much not random, but

are tied to the Greenland delta O-18 temperature proxy with variability only in amplitude and initial temperature. Please reword.

*The sentence has been reworded to "$\delta^{18}O$ data is used as a prior on the shape of the thermal histories, but we leave two scaling parameters free – minimum temperature at 20 ka, and amplitude (temperature difference between at 20 ka and present)– and we did not include the role of ice on setting the rock temperature while it was ice covred."*

Discussions generally: I won't comment much on these inferences about past climate systems at this stage, because I think it will be very important to first demonstrate model success in capturing simple variations within a periodic forcing model, which, at this stage, has not been done.

---

## Author Response (AR1)

Please note that the reviewers' comments are in black and responses are in blue.

**Editor**

Comments to the Author:

Dear Dr. Biswas and co-authors,

Thank you for your clear responses to the referee comments on your Discussion manuscript and your willingness to consider their suggestions. I invite you to submit a revised manuscript in the spirit of your responses. Because the nature of the referee comments and your proposed manuscript changes are fairly technical, any revised manuscript will go through an additional round of peer-review (this additional round will not be open to the public, though eventually the reviews and responses become part of the public record). This requires that I indicate a decision of "major revisions" on this manuscript.

Sincerely,

Alberto Reyes (handling editor)

Non-public comments to the Author:

Just a few additional comments based on the reviews and my own reading of the manuscript:

1. I suggest rephrasing part of the paragraph beginning 4/30-31 of the original manuscript. Climate is a much longer term phenomenon than daily/season changes in conditions. And on longer timescales, climate does not follow Milankovitch forcing in a linear way but rather is paced by it. I realize that this section represents a theoretical exercise, but the way the section starts it would be easy for readers to get the wrong impression.

We have changed the sentence to "*Earth's climate varies in cyclical way, at multiple time scales from years to decades, centuries, and millennia, influenced by periodic variations in the Earth's orbit, known as Milankovitch cycles, at 25.77, 41 and 100 kyr*". See page 4, line 28-29 in the revised manuscript.

2. The first and second substantive comments from Rev 2 involve some potential misunderstanding of assumptions made in your study. I encourage you to provide additional clarity on these points in your revision.

Please see the replies to those comments the action taken in the manuscript.

**Anonymous Referee #1**

General comments:

The authors are investigating an interesting question–whether luminescence signals from K-feldspars in bedrock might archive changes in recent temperatures at Earth's surface. More specifically, they ask whether we can resolve recent changes in temperature periodicity. This question is important and worth pursuing.

I commend the authors for the layout of this study. Their approach involving sensitivity analyses, calibration to sample specific kinetics and attention to climatic complexity have resulted in an interesting manuscript with potential for significant scientific impact.

We appreciate the reviewer for discerning the potential of this work.

However, the present work needs significant clarification and expansion to yield a robust estimate of past temperatures. Specifically, the authors must determine how changes in amplitude, period, and mean temperature influence luminescence signal growth and depletion in a holistic way. Currently, the treatment is partial. Once this is done, the authors should give a more direct comparison of actual and predicted temperature histories in order for the reader to better examine the predictive success of the model.

Please see the replies below.

These points and others are detailed below.

1/22: "Earth's climate fluctuates in a cyclic way" While there are many internal cycles to climate systems, this characterisation might be too simplistic, especially at the timescales involved here ($\sim 10^1$ - $10^2$ kyr), where abrupt periods of change are common. Temperature changes during the Holocene, for example, can hardly be approximated as cyclic.

We have changed the sentence to… "*Earth's climate fluctuates in a cyclic way, from years to million-year time scales driven by Earth's orbital processes and rare aberrant shift and extreme climate transients during the last $10^3$ to$10^5$ years*". See page 1, line 22-23 in the revised manuscript.

1/35: "equivalent diffusion temperature that is always higher than the actual mean temperature" This statement, while true, gives the false impression that this is an intractable bias. For a system with well-characterised diffusion kinetics, the relationship between a given temperature history (e.g., a forward model) and the EDT is well known. In other words, paleothermometry using the He-3 paleothermometry technique must rely upon comparisons against prescribed temperature histories in the same way as paleothermometry with luminescence techniques. This is not a comparative disadvantage of the noble gas technique, but a similar limitation as faced in the current study.

This is true. We meant that a single thermometric system (OSL or $^3$He) will always provide a single equivalent temperature (like EDT in $^3$He) for a complex thermal history whereas a multi thermometric

system (here TL) with different temperature and time sensitivities will have the potential to infer a more complex thermal history. We have clarified the sentences to *"Tremblay et al. (2014a, b) recently introduced a new paleotemperature proxy based on the thermal stability and release of $^3$He and $^{21}$Ne noble gases in quartz. The system has a single energy level and is therefore only able to estimate a single equivalent diffusion temperature (EDT). As a result, reconstruction of complex rock temperature histories challenging."* See page 1, line 33-36 in the revised manuscript.

2/6: The distinction between thermochronology and paleothermometry is not entirely clear in the language of this study. If what you aim to resolve is the temporal variation of temperature through time, you are describing thermochronology. If instead, you mean to resolve a past temperature which is representative of some time period (the measurement of which will be affected by seasonal variability and so on), then what you are describing is paleothermometry. I would encourage more precision when you describe these concepts.

Paleothermometry is a methodology for determining past temperature, and thermochronology is the study of the thermal evolution of rocks, and they both rely on the same principle. However, thermochronology is more commonly used for rock cooling related to exhumation. So to avoid confusion, we now use paleothermometry throughout the manuscript. We only refer to thermochronometry when we refer to previous work, which was developed for thermal evolution of rocks associated with exhumation.

 2/12-21: Please add corresponding references for these observations.

Amended. See page 2, line 21 in the revised manuscript.

2/36ff: The physical meaning of this model is unclear. This is obviously of fundamental importance, as the kinetic model that is chosen will determine all predictions of past thermal history.

From Eq. 1, it would seem that the authors expect to model some number of individual traps, each with a singular values for D0, E, s, s-tilde, rho', a, and b. All of these traps are modelled as disconnected.

And yet, the transition in parameter values from one measurement temperature bin to another is smooth. This is true for D0, for rho', for E, for s, and for b within Fig. 1. This observation strongly suggests continuity in the underlying kinetics, not only for trap depth(s) but for the system as a whole. To fit each measurement bin as a separate and disconnected trap seems suspect. A unified treatment would be preferable.

Several studies suggest that broad TL glow curve from feldspar arises from a continuous distribution of trapping energies, which is suggested by several methods, like $T_m$-$T_{stop}$, the initial rise method, and analysis of fractional glow curves (Biswas et al., 2018; Duller, 1997; Grün and Packman, 1994; Pagonis et al., 2014; Strickertsson, 1985). Regardless, it is difficult to isolate a single trap with distinct kinetic parameters.  Instead we assign the most probable kinetic parameters for each thermometer (glow curve temperature) along the TL glow curve and in this manner we determine kinetic values in a continuous, rather than in a disconnected manner. This is the method that we have adopted here and in Biswas et al.

(2018). We then arbitrarily choose 10 °C TL temperature windows as distinct thermometers. A continuous distribution of trapping energies can be assumed as the sum of a large number of discrete traps (Pagonis et al. 2014). Thus a continuous distribution of trapping energies is discretized as shown in the figure below. This has been added in the supplementary materials (section S1.2).

[Figure]

**Fig. S1: The evaluated continuous distribution of trap depth (E) of sample MBTP1 (circles with error bar) and its discretization in 10 °C windows (green box; width is 10 °C and height 5 % of the median value).**

Another issue to address is whether the same recombination centers are accessed by this distribution of traps during athermal fading. If so, as would seem unavoidable to some degree, rho should be kept constant (the density of centers being a property of the material). rho' can then be related to the underlying activation energy via the alpha term. This should be attempted for internal consistency.

Here we only use $\rho'$, which includes alpha as $\rho' = \frac{4\pi\rho}{3\alpha^3}$. Alpha controls the rate of fading through the lifetime $\tau = s^{-1}\exp(\alpha r)$. It is expected that with increasing activation energy fading rate ($\rho'$) should decreases as the tunnelling depth increases. Indeed we get similar relationship between fading rate ($\rho'$) and activation energy (Table 1) as shown in figure below.

[Figure]

**Fig. X: Plot of distance dependent fading rate ($\rho'$) and activation energy ($E$) of five thermometers (200-250 °C, 10 °C interval) of sample MBTP9.**

According to the second term on the RHS of Eq. 1, it seems that the authors model thermally-activated recombination locally (since the term is dependent upon the nearest neighbor distribution). If so, then observations of signal loss at room temperature could also be caused by this pathway. This deserves comment.

Thermal loss of the TL signals which is a more diffusive process (athermal) can occur through the conduction band. However, the power term $b$, accounts for the nonlinearity (delayed) arise due to presence of band-tail states. However, athermal fading is known as a tunnelling process. Separate treatments of thermal and athermal loss has adopted successfully in several previous studies (Guralnik et al. 2015a for IRSL of feldspar; Biswas et al. 2018 for TL of feldspar).

3/29-30: Can we be confident that the kinetic parameters pertain to geologic timescales? Specifically, is there good evidence or reason to think that mixed order kinetics are predicted at low temperatures and long timescales (natural) as well as high temperatures and short timescales (lab)? Competition effects, for example, could easily produce observations of b > 1 for lab measurements whereas the concentration of charge activated on natural timescales would be orders of magnitude smaller.

This is a difficult question and can never be answered in true sense but the premises of the approach used here was validated in Biswas et al. (2018), by successful recovering temperatures experienced by rocks in the KTB borehole, which have been in thermal steady state for several millions of years. We have included that statement in the manuscript.

4/19 and Fig. 3: I find this figure a little difficult to interpret. In particular, the way that you have defined 'memory time' should be a bit clearer. If I've understood the meaning of this metric correctly, one suggestion would be to compare everything against 'tchange = 100ka' or to extend the x-axis to include 't-change=150ka.' By doing this, you could then visually show the meaning of the 't-memory' by comparing two horizontal lines, one at t-change=150ka (or 100ka, depending), and the other at the asterisk height. You could then annotate this difference as 20%.

Amended as recommended. In the figure the 20% changes have been demonstrated clearly and text has been added in the figure title (Fig. 3).

It would be good to consider also the influence of measurement uncertainty. Accurately resolving the difference between [n-bar] = 2.0e-3 and 2.4e-3 would likely involve a lot more relative uncertainty than discerning between, say, [n-bar] = 0.5 and 0.6.

Typical photon counts of the present sample for near saturation ($\bar{n} = 1$) is >$10^5$ counts per second (cps), our instrumentation can resolve a few hundreds of cps with 20% uncertainty ($\bar{n}$ of the order of $10^{-3}$). This is discussed in Section 4.3 (Page 8, line 23-26) as "*Typically, the minimum detectable limit for the present instrument is ~ 300 photon counts per second (cps) considering the signal should be three times of background level which is ~ 100 cps. The present highly luminescent feldspar has maximum photon count of ~$10^6$ cps. This restrict to use the TL signals up to ~$10^{-3}$ % of maximum TL signals*"

4/30-31: '10 - 100 kyr timescales' This should be a bit more specific. During the Quaternary, the 100kyr and 41kyr periods were most dominant (e.g., Raymo et al., 2006; Hinnov, 2013).

We have changed the sentence to "*Earth's climate varies in cyclical way, at multiple time scales from years to decades, centuries, and millennia, influenced by periodic variations in the Earth's orbit, known as Milankovitch cycles, at 25.77, 41 and 100 kyr*". See page 4, line 30-31 in the revised manuscript.

Fig. 4: This is a nice figure. Please adjust so that the 'thermometer' labels correspond to each panel. For example, in panels j-l, if the reader uses the labels on the far righthand side, they might conclude that the bottom-most series in panel k refers to the 220-230C chronometer.

Thanks. Amended as recommended (Fig. 4).

5/7: 'implies a gradient' I would say this behavior more accurately 'results' from a gradient in thermal stability, given that the kinetic parameters are imposed and known.

Amended.

5/9: 'the thermal stability decreases with increasing temperature.' I disagree with this statement. The thermal stability for a given TL thermometer is known and fixed in your setup. So it is not the stability that is changing, but rather the probability of detrapping.

Thanks for pointing this out. The sentence has been reworded to "*This is because the probability of detrapping increases with increasing temperature*". See page 5, line 7 in the revised manuscript.

5/10: 'the higher temperature TL thermometers remain relatively insensitive to such periodic temperature forcing.' This is misleading. The insensitivity of the higher-T thermometers reflects the mean temperature values that you have chosen. If the temperature oscillated about a higher mean value, the same periodic filling/emptying behavior would be seen with the high-T thermometers. This is evident in Fig. 4 panels j-l, where different thermometers oscillate with comparable magnitude when a range of mean temperatures are tested.

*The sentence has been changed to "Finally, the higher temperature TL thermometers (near to 300 °C) remain relatively insensitive to such periodic temperature forcing ($T_{mean}$ up to 30 °C); with increasing $T_{mean}$ the higher temperature TL thermometers become more responsive.". See page 5, line 7-9 in the revised manuscript.*

5/13: '10 ka to 1 Ba' If Ba represents 'billion years,' please change to 'Ga.'

*Amended.*

5/15: 'For P « tau' This comparison must be qualified. The lifetime (tau) will depend on a chosen, singular temperature value. Choosing a singular temperature value for oscillating temperature requires some simplification that is not described (e.g., mean temperature? EDT?). Please clarify this issue.

*The sentence has been clarified to "For $P << \tau$ (e.g. Fig. 4d where P=1 ka and $\tau$ spans ~10 ka to 1 Ga for the 200 to 300 °C TL thermometers for $T_{mean}= 0$ °C), the value of $\bar{n}_{osc}$ exhibits small fluctuations but always remains lower than $\bar{n}_{iso}$". See page 5, line 14-16 in the revised manuscript.*

5/16: 'This result implies that smaller periods (<1 ka) do not influence trapped charge equilibrium levels in an oscillating fashion and cannot be differentiated from the trapped charge population resulting from an isothermal condition.'

This statement is incomplete and only conditionally true. For argument's sake, assume that the 200C TL thermometer has a lifetime of 10 ka at 0C. If the ambient temperature oscillated with an arbitrarily large amplitude (say 100C to make the point obvious) but with a period of only 1 ka or 100 yr, you would find that the fractional saturation would oscillate in response to the temperature forcing, depleting completely and then partially regenerating.

This won't happen if the temperature period is much smaller than the growth timescale (D0/Ddot). If that is the case, then the thermal imprint upon the sample approaches a steady value determined by the maximum temperature experienced.

Temperature amplitude and the relationship between the forcing period and sample growth timescale both matter. To make this comparison between lifetime and period, these factors must be incorporated.

*The sentence has been modified to "This result implies that smaller periods (<1 kyr) and Tmean (<30 °C) do not influence trapped charge equilibrium levels of 200 to 300 °C TL thermometers in an oscillating fashion and cannot be differentiated from the trapped charge population resulting from an isothermal condition. We must mention here, if the amplitude of oscillation increases the oscillating*

*response to trapped charge equilibrium levels will be relatively prominent.*". See page 5, line 16-19 in the revised manuscript. The effect of oscillating parameters, mean temperature, amplitude of oscillation and period on trapped charge population is discussed is section 2.2.2 (Page 5, line 28-35) and shown in Fig. 5.

5/19: 'remains correlated' and 'deviates from...temperature forcing' The meanings of these statements are unclear. n-bar behavior is distinct between isothermal and oscillating temperature histories for all periods; it is not an issue of matching and not matching, except for the highest-temperature systems, which are insensitive to the temperatures prescribed here. Please be more specific with these observations and, following from the previous comment, please do incorporate growth timescales, as these are of obvious relevance here.

Clarifications have been made. The sentence has been changed to "*Similarly, the present day $\bar{n}_{osc}$ remains indistinguishable from $\bar{n}_{iso}$ when $P>>\tau$ (e.g. see the behaviour of the low temperature TL thermometer shown in Fig. 4l)*". See page 5, line 21-22 in the revised manuscript.

5/24: 'Therefore, temperature variations can be reconstructed...' Just to reiterate, you must demonstrate the complex relationship among mean temperature, temperature amplitude, trap stability and regenerative timescales before attempting to reconstruct temperature variability. Additionally, what has been shown in Fig. 4 is that, for a given amplitude, different periods leave different imprints upon the shown thermometers. You have not yet demonstrated that you can accurately reconstruct differences in variability. Moreover, the results from Fig. 5 (panels b, c) seem to indicate that you cannot easily differentiate between various amplitudes or periods.

This is discussed in section 2.2.2 and Fig. 5 where we show that the present day trapped charge population ($\bar{n}_{present}$) is highly sensitive to the mean temperature ($T_{mean}$) and less sensitive to the amplitude of oscillation ($T_{amp}$) and period ($P$). Moreover, we also mention that "*Although the $\bar{n}_{present}$ is less sensitive to the amplitude and the period, the pattern of $\bar{n}_{present}$ of different thermometers is unique. This ensures that complex thermal histories comprising mutiple harmonics with periods of about tens of kyr, but distinct from one another, can be recontructed*". See page 5, line 32-35 in the revised manuscript. Additionally, now we have added a new synthetic test to emphasis the sensitivity of the three periodic parameter ($T_{mean}$, $T_{amp}$ and $P$) on the present day trapped charge population ($\bar{n}_{present}$) in section 3.3 and results are shown in Fig. 6.

5/31-33: 'This ensures that complex thermal histories...can be reconstructed.' Following from the previous comment, this is not yet demonstrated.

See the answer above.

6/11: 'considering that the other parameters are identical.' Unclear what this means. From Fig. 1, it would seem that the kinetic parameters other than the thermal parameters (E, s) vary between the thermometers. Or do you mean something else?

We agree this was confusing. We have now removed these unnecessary words.

Fig. 7: This is not the most informative way to show your predictive ability. Unlike with thermochronology studies, where the T-t path is really the predicted feature, what you are more accurately doing is predicting the temperature minimum and amplitude (e.g., ll. 5/24-29). So, it would be much more informative to see these values, actual and predicted.

It is true that the only difference with the inversion of thermochronometric data is that we impose the shape of $\delta^{18}$O. To address the reviewer's comment we have added histograms of the two main scaling parameters: the minimum temperature $T_{base}$ (temperature at 20 ka) and the present temperature, $T_{present}$ (which is $T_{base}+T_{amp}$ as shown in supplement Eq. S9). See Fig. 7.

Additionally, the current figure makes it appear as if you are able to resolve the fine structure of the T-t series, which of course you are not.

Indeed, we cannot. Rather this figure shows the scaling of the $\delta^{18}$O curve. The reply to the previous comment should have addressed this comment.

7/16-18: '[Fig. 7a,b,c shows] it is possible to recover all three thermal histories within the 1-sigma confidence level.' My previous comment will apply here as well. What matters for this experiment is the degree to which you are able to predict amplitude and minimum.

To demonstrate that you could recover an arbitrary thermal history well, you would need a different test.

Please see the reply of previous comment. We clearly mention we need prior information for the pattern of the thermal history ($\delta^{18}$O), as typically done for inversions, but that we can constrain the temperature minimum and amplitude. Please see section 3.3 and 3.4 in the revised manuscript.

8/7: 'This restrict...' Grammar

Amended.

8/15: This dose range, with upper doses at 0.9 and 1.9 kGy may not be sufficient to observe saturation in K-feldspar TL. Please demonstrate that these signals are saturating or add greater doses to better constrain lab saturation intensity.

1.9 kGy is sufficiently high as it is greater than $2D_0$; maximum $D_0$ is less than 800 Gy (see Table 1).

8/33: 'no effect for doses below 100 Gy' Wouldn't the far more important question be whether there is sensitivity change above 100 Gy? After all, the majority of given doses are above 100 Gy and these responses allow you to determine the saturation values.

The sensitivity change in the NCF method permits determination of the equivalent dose correctly by correcting for possible sensitivity changes during a sample read out and is generally tested on low doses (see Singhvi et al. 2011).

9/13-15: 'The rationale here is that all temperatures follow the delta O-18 data, but the amplitude...and mean temperature are unknown.' Please make clear that this is a stated assumption and not an inference from studies (unless it is, in which case state that). I think is not obvious that the climate signal on Mont Blanc would mirror a Greenland ice core signal, so this assumption probably warrants justification.

To clarify, the text has been modified as follows "*We assume that the atmospheric temperatures of the Mont Blanc massif followed the trend observed for the Greenland ice core data over the last 60 ka. Note that temperature increase during the last glacial cycle was synchronous with the temperature anomalies observed in Greenland (e.g., Heiri et al., 2014; Schwander et al., 2000; van Raden et al. 2013). The rationale here is that all temperatures in the Mont Blanc massif follow the Greenland ice core δ18O data but the amplitude of temperature oscillation (minimum temperature at ~20 ka to maximum temperature at the present day) and mean temperature are unknown.*". See page 9 line 31-36 in the revised manuscript.

Fig. 9: Is Fig. 9 referenced in the main text?

Now it is mentioned.

Fig. 9 and 10: As with Fig. 7, please recast to compare the predicted and actual values for the amplitude and base temperatures as these are the variables being investigated.

We have now added subplots (histograms) of the two parameters, temperature minimum which we define as $T_{base}$ (temperature at 20 ka) and present temperature, $T_{present}$ (which is $T_{base}+T_{amp}$ as shown in supplement Eq. S9). See Fig. 9 and 10.

9/28: 'can constrain thermal history of ~50kyr.' I do not think this has been demonstrated yet, as an extension of my comments regarding pg. 5.

In Fig. 3 we show that $t_{memory}$ can be up to 50 kyr for typical temperature of 10-20 °C. If the amplitude of temperature change is higher, the method will be sensitive, thereby improving our ability to constrain the thermal history constraining of thermal history will be more precise. See section 2.2.1.

Also, unclear what 'A higher temperature fluctuation' means and whether you've actually shown this.

This is shown in Fig. 3.

10/14: 'At those depths [of 7 and 62 cm], mean temperature should be constant.' Certainly, there will be seasonal temperature variability at a depth of 7 cm. Is this what you meant to say?

We mean that at depths of 7 and 62 cm the rock temperature will be equilibrium with the atmospheric temperature. The sentence has been modified to "*At those depths, mean temperature should be in equilibrium with atmospheric temperature (e.g. Hasler et al. 2011)*" . See page 11, line 10-11 in the revised manuscript.

10/17: For inverse modeling of a natural sample, the time-temperature histories were completely random' If I have understood the previous text correctly, the histories are very much not random, but are tied to the Greenland delta O-18 temperature proxy with variability only in amplitude and initial temperature. Please reword.

The sentence has been reworded to "*For the inverse modeling of natural samples, $\delta^{18}O$ data are used as a prior on the shape of the thermal histories, but we leave two scaling parameters free – minimum temperature at 20 ka, and amplitude (temperature difference between at 20 ka and present)– and we did not include the role of ice on setting the rock temperature during glaciation.*". See page 11, line 13-15 in the revised manuscript.

Discussions generally: I won't comment much on these inferences about past climate systems at this stage, because I think it will be very important to first demonstrate model success in capturing simple variations within a periodic forcing model, which, at this stage, has not been done.

Anonymous Referee #2

This manuscript proposes a new method to study/reconstruct paleo-temperature using TL of feldspar from rock surface. Reconstructing paleo-temperature is an important topic in climatic change study, so the attempt of this study is important and worth for publication. However, there are some technical issues that are not settled well, which prevents convincing me that this is achievable. Here I summarise my major concerns.

1) Kinetic model: the authors considered three processes in their model, including dosing (growth), thermal decay and athermal decay (fading), which are represented by different terms in their equation 1. They then estimated the parameters based on TL measurements in different ways. For both the growth curve parameters (D0 and a) and fading parameters (rho'), my understanding is that they were based on the signals from different integrals of the TL glow curve (e.g., 200 – 250 C in 10C interval). However, for the thermal decay parameters (E, s and b), they used a Tm-Tstop method, in which the signal from each temperature interval is obtained from subtracting consecutive fractional glow curve. That means, the signals they used to estimate present-day charge population ($\bar{n}$), growth curve and fading are based on simple integral of TL signals at different intervals, which obviously are a mix of signals from a range of trapping energy levels, but the thermal kinetic parameters are based on single (or narrow-range) trapping energy levels. That says, the authors did not separate the TL signals for constructing their model using a similar way (Tm-Tstop method) that they did for estimating the thermal decay parameters. This is problematic, as the combination of different trapping levels are not linear, so their model (equation 1) simply becomes invalid when the signals being analysed are associated with a range of trapping levels are analysed. One need to makes sure that the different parameters in Equation 1 are all obtained from the same signal associated to a single or narrow-range energy level. I am not use if this can be achieved as the combination of MAR protocol (and fading test) and Tm-Tstop would be very difficult to achieve.

We do not assume that the thermal kinetic parameters are estimated on single trapping energy levels, as the number of traps are not known. In the absence of any information of the number of peaks it difficult to deconvolve the TL glow curve. So instead we use a 10 °C window and treat them as different 'thermometers', rather than single trapping energy levels. (Note that the rationale behind choosing a 10 °C window is the uncertainty of sample temperature during TL measurement in the laboratory.)

We constrain the growth parameters ($D_0$ and a) and fading parameter ($\rho'$) for these different 10 °C temperature windows. To evaluate the thermal decay parameters ($E$, $s$, $b$) of each 'thermometer', we need to know the distribution of these parameters along the TL glow curve and find the most probable decay parameters of different thermometers from the distributions. Pagonis et al. (2014) demonstrated that the distribution of thermal decay parameters along a TL glow curve can be obtained using the *Tm-Tstop* method, subtracting the fractional glow curve and fitting with a TL equation. Biswas et al. (2018) adopted this method for thermochronology and showed it could be used to extrapolate laboratory data to geological timescales, as a proof-of-concept.

The subtraction of consecutive fractional TL glow curves, obtained through the *Tm-Tstop* method, and fitting of the subpeak provides the characteristics (kinetic parameters) at that TL temperature. The

integrated TL signal is a mix of different signals but the major contribution comes from the subpeak at that temperature range and a small fraction from the subpeaks either side of the temperature. Biswas et al. (2018) demonstrated (Fig. S4 in their supplement) that the life time distribution along the TL glow curve (present method) exactly match with classical lifetimes of the main dosimetric peaks of feldspar (as summarised in Aitken's TL book, 1985, Table E.1). This ensures the estimation of kinetic parameters of integrated TL signals from a distribution holds well.

Since last six to seven years there are lot of development on kinetic of luminescence of feldspar. The theoretical model for the rate equation of trapped charge population in feldspar has been described in several ways; first order kinetics (Brown and Rhodes, 2017; Yukihara et al., 2018), general order kinetics (Biswas et al., 2018; Guralnik et al., 2015b), charge transport through sub-conduction band-tail states (King et al., 2016a; Li and Li, 2013), Gausian distribution of trapped energies (Lambert et al., In Revision), or localized recombination in randomly distributed defects (Jain et al., 2012). What is common to all these models is that luminescence of feldspar is complicated and exhibit a non-linear non-first order kind of behaviour due to either presence of sub-conduction band-tail states (Morthekai et al., 2019; Poolton et al., 2002) or complex charge transport mechanism. TL in feldspar is even more complicated because it shows continuous distribution of trapping energies (Biswas et al., 2018; Duller, 1997; Grün and Packman, 1994; Pagonis et al., 2014; Strickertsson, 1985) and TL is a more diffusive process than OSL; OSL of feldspar has resonant energy levels (Hütt et al., 1988). Different models were reviewed and tested by Guralnik et al. (2015b) who suggested that the general order kinetic, a mathematically simplified model, could be used to explain luminescence phenomenon well. We adopted this model for TL of feldspar where the power terms ($a$, $b$) accounts for the nonlinearity involved in the TL of feldspar. The efficacy of using general order kinetics has been demonstrated to samples with known thermal history (KTB borehole samples) for OSL of feldspar (Guralnik et al., 2015a) and TL of feldspar (Biswas et al., 2018). This section has been added in the beginning of discussion section, page 10, line 7-19.

2) The authors simply assuming feldspar consist a continuous trapping energy level. However, it has been commonly accepted that band-tail states play important role in the luminescence process (including TL and IRSL) in feldspar (Poolton et al., 2002). I do not see any reason to discard the band-tail states from their model.

There is ample evidence from the literature suggesting that feldspar consists of continuous trapping energy levels (e.g. Biswas et al., 2018; Duller, 1997; Grün and Packman, 1994; Pagonis et al., 2014; Strickertsson, 1985).

We do not discard band-tail states in feldspar. See the reply of previous comment. Feldspar luminescence is complicated and several models that describe thermal detrapping processes in feldspar have been proposed (see summary in Gurlanik et al., 2015b). Moreover, the TL of feldspar is a more diffusive process than IRSL which excites a resonant energy level below conduction band from which electrons either tunnel to recombination centers or hop through the band-tail states and recombine. The general order kinetic model is the simplest model available and the power term b (>1) accounts for the non-linearity that arises due to the presence of the band tail states (restricted mobility) or other complex

process. This model has been used for a long time and most recently was used by Guralnik et al. (2015) for IRSL of feldspar and Biswas et al. (2018) for TL of feldspar.

3) How sensitive is the model to dose rate ($\dot{D}$)? The dose rate would play an important role in filling the traps during natural process, so I would expect that it will somehow influence the model results. Unfortunately, the dosimetry of the samples is poorly described. How did the author estimate the dose rate of K-feldspar? The author appears to simply crush the rock and select 150 –250 um grain size range. What are the original grain sizes of the K-feldspar minerals in the rock? Did the authors make any rock slide to investigate this? This is critical as there are a large contribution of internal dose rates for Kfeldspar.

The dose rate values were taken from Lehman et al. (2020). It is mentioned in title of Table 1. They mention in their paper (supplement) as follows: "Environmental dose rates were calculated using DRAC online calculator (Durcan et al., 2015), assuming a grain size between 750 and 1000 μm and water content of 2%". The grain sizes were estimated through thin section analysis by Lehman et al. (2020).

4) The authors applied the NCF method to overcome sensitivity change issue for TL measurements. They do realise the limitation of this method as it is based on extrapolation of the NCF values from low temperature to high temperature region, which is unreliable. Although the authors tested the effect of initial sensitivity changes on the modelling results and found very little changes for their sample, it does not guarantee that this applies to other samples and situations. The reason that the sensitivity changes did not affect the results is simply because that their samples are young and the growth of signal still lies on the linear part of growth curve, so any systematic changes in the sensitivity result in a proportional changes of different signal integrals. For older samples or high-De samples, however, this may result in non-proportional changes among different signal integrals (because the different D0 values for different integrals), and, hence, different model results in paleo-temperature. This potential problems should be appropriately acknowledged as at present it gives false impression that the initial sensitivity change does not matter.

We agree and have included appropriate caveats. Sensitivity change is a limitation for TL measurements and as such we carefully screened the data. As for all TL measurements this will need to be done in further studies but as this is well known we do not emphasise the point here, but rather give a detailed account of how we addressed this challenge.

To circumvent the initial sensitivity change the NCF was introduced (Singhvi et al. 2011) as shown in Fig. 8a. The limitation of this method is that the NCF can only be measured for the lower temperature part of the TL signal (<200 °C) where our region of interest (ROI) is 210-250 °C TL. In the absence of a strong correlation (dose response) between low temperature TL and the ROI, we extrapolated. As the reviewer notes, this approach is appropriate for our sample suite. See section 4.3.1.

Minor comments:

5) Line 21: Credits should be given to Li and Li (Li and Li, 2012) who firstly proposed the idea of multiple-thermometers using TL (although not implemented in their study), and they also first

introduced the rate equation to investigate the effect of single growth and saturation on OSLthermochronology.

Reference added.

6) Figure 1: The authors should at least provide some typical TL glow curves for their samples before showing the kinetic results.

Amended.

7) Figure 1c: The fading parameter (rho') shows systematic change as a function of temperature up to 280C, but it suddenly become 'no fading'. This is surprising. The different integral signals represent a continuous mixture of signals of different athermal features, why one can obtain a sudden change in the fading? Is it because that the fading rate has large uncertainty range consistent with zero fading? In this case, it would be problematic to say 'no fading', as statistically it is also like to be 'fading'.

Beyond 270 °C, the fading parameter ($\rho'$) is less than $10^{-10}$ with large uncertainty and is considered to be a non-fading signal. Even a fading curve with $\rho'<10^{-7}$ looks parallel to the time axis.

8) Figure 8b: Why not plot the results for other temperature range (e.g., 120 – 150 C)?

The objective is to see whether $NCF_{100}$ is dose dependent or not.

9) Figure 8c: what are the errors for the NCF at high temperature range (200 – 250C)? Have you incorporated the NFC errors into the final results?

The errors for the NCF at the high temperature range (200 – 250 °C) are stated in Table 1. We did not incorporate NFC errors into the final results.

10) Table 1: Why there are no errors for E, s and b? Why an arbitrary error of 5% is assumed, rather than their actual analytical errors?

See Fig. 1. The distribution of E, s and b are a bit scattered and we spline fit the trend. So we did not calculate the regression of the trend and rather assumed an working error of 5%. This is mentioned in the caption of Table 1.

11) Figure S3: why there is only one natural point but 3 regenerative points for each thermometer? Did the author just measure one aliquot for natural?

The natural point is the average of three discs. This is mentioned in the title of Fig. S3.

**List of corrections**

1. We have addressed all the queries of the reviewers and incorporated them in the revised manuscript and actions taken are mentioned in the replies to the reviewers' comments.

2. Two major changes are made in the this revised version of manuscript, 1) we introduce a new synthetic experiment in section 3 to justify the rationale behind using the trend of $\delta^{18}O$ in the inverse modelling. 2) we have added a paragraph in the beginning of Discussion section to justify the model used.

3. All changes made in the manuscript can be seen in marked-up manuscript version (track change mode) attached herewith.

[revised manuscript text omitted]

---

## Author Response (AR2)

Please note that the reviewers' comments are in black and responses are in blue.

**Editor's comment**

I have now received an assessment of your revised manuscript from Reviewer 1 of your original submission. Overall the reviewer is satisfied with your revisions, and I agree with their recommendation that the manuscript be published in Climate of the Past following minor revisions to clear up a few remaining points.

Thanks for agreeing with the potential of the paper.

The reviewer also points out that some criticisms remain in terms of your approach to dealing with the kinetics of feldspar thermoluminescence. In light of this, I'd encourage you to take one more look at the related critical comments of Reviewer 2 (from the original manuscript) and see if there are any appropriate spots to further acknowledge some of the uncertainties/ambiguities in your approach.

We have carefully gone through the model used here. As mentioned in the text the model is based on the previously published paper (Biswas et al., 2018, EPSL) on TL thermochronology. The model was tested on KTB borehole samples which have been in thermal steady state for several millions of years and to samples from Namche Barwa where results were corroborated with other methods, like OSL and A-He. However, like any other method, there is always scope of improvement, and we are open for it.

Please include with your revised manuscript a response letter that highlights any changes you've made, or that justifies why you don't feel a change in response to a particular suggestion is warranted. I look forward to seeing this work published in Climate of the Past.

Please see the replies to the reviewer's comments and action taken below. And a marked-up version of manuscript is attached herewith.

**Reviewer's comments**

The authors have done a thorough job of clarifying most of the issues I introduced in my initial review. The revised manuscript is easier to follow and more impactful.

Thanks.

In keeping with my earlier review (and in agreement with the other reviewer's two initial points) I view that the kinetics underlying feldspar TL are still ambiguous and that the approach taken here, of treating temperature bin ranges as independent thermochronometers with isolated kinetic parameters, is not ideal. Like the other reviewer rightly pointed out, the compound nature of the feldspar glow curve implies that even the narrowest bin comprises emissions of multiple stabilities. That said, a satisfactory

unified model for feldspar TL isn't agreed upon yet, and the authors have made clear their methodology, so I think their approach is acceptable.

*We acknowledge the reviewer's view on Feldspar TL model that there is no unified model agreed by all TL users. However as mentioned above that the model and approach used here is based on the previously published paper (Biswas et al., 2018, EPSL) on TL thermochronology. The model was tested to KTB borehole samples which have been in thermal steady state for several millions of years and to samples from Namche Barwa where results were corroborated with other methods, like OSL and A-He.*

I have a few outstanding comments listed below, but in general I think this study is in good shape and should prove to be an valuable resource for the community.

Specific comments:

2/36ff:

To my earlier comment that it would be neat to relate activation energy to rho' via the alpha term, what I meant was that the same rho (dimensional) should pertain and you can use

alpha = 2*sqrt(2*[mE^*]*E)/hbar

to evaluate the value of rho (dimensional), since E and rho' are known for all bins and all other values are fixed and known.

*This we have answered previously. However, we have added a sentence "It can be noted that with increasing the TL temperature (or thermometer), the activation energy ($E$) increase and athermal fading ($\rho'$) decreases (Table 1)". Please see Page 9, line 23-24.*

To my comment about thermal loss at room temperature, I understand that previous studies have dedicated separate terms to thermal and athermal loss components, but my question was whether signal loss at RT is negligible, assuming only the 'thermal loss' term. It would be simple to test this by simulation by using the best fit parameter values, populating the traps, turning off the athermal term and seeing whether the signal decreases at T=20C.

If this turns out to be negligible, then all is fine, but if loss does occur, the fading data should be evaluated in terms of both loss pathways simultaneously.

*We have mentioned in the measurement details (Section 4.3.1) that for athermal fading aliquots were preheated to 200 °C prior to storage. A cut-heat up to 200 °C remove all the trapped electrons that are unstable at room temperature over laboratory time scale (maximum few days or even year). The resident time of the trapped electron corresponding to 210 °C TL signal (lowest thermometer) at room temperature is of the order of hundreds of years (Biswas et al., 2018). The 210 °C TL is sensitive to room temperature (20 °C) over geological time scale but not over laboratory storage time during the*

fading measurement which is few days only. Thus the loss of signal during fading measurement is purely through athermal pathway not through thermal loss.

10/14: The revised statement that "At those depths [of 7 and 62 cm] the rock temperature will be at equilibrium with the atmospheric temperature (Hasler et al., 2011)" is still problematic and in conflict with the cited paper. For example, Fig. 8 in Hasler et al. (2011) shows a strong temperature gradient for many sites down to ~1 m.

Likewise, the Magnin paper shows a clear and pronounced temperature gradient extending beyond 2 m, with the exception of the cracked rock, which is cooled by ventilation.

So, a sample taken from 7 or 62 cm will likely not record the same temperature as the atmospheric temp. This is a difficult problem and one that doesn't necessarily need to be solved here, but the text should be straightforward about this complication.

We agree to this point. Both Hasler et al. (2011) and Magnin et al. (2017) shows some temperature gradient with depth with the exception of the cracked rock, which is cooled by ventilation. However, Magnin et al. (2017) shows it is nearly constant (or maximum a difference of 1 °C between surface and 2 m depth) for averaging over annual time step (Fig. 6c of Magnin et al. 2017). The maximum erosion depths, 7 and 62 cm, for the two samples were calculated from the erosion rates ($3.5 \times 10^{-3}$ and $3.2 \times 10^{-2}$ mm/yr) and maximum erosion duration (20.9 and 19.5 ka) from Lehman et al. (2020). Thus these slow erosion constantly changing the temperature of the target material (present day surface sample) but with a very lower extent (maximum up to 1 °C in present case). Thus 
[revised manuscript text omitted]

|---|---|---|---|---|---|---|---|---|---|---|---|
| MBTP1 | 210-220 | | 766 ± 51 | 1.00 ± 0.09 | 1.24 | 11.62 | 1.46 | -6.02 ± 0.08 | 0.17 ± 0.03 | 1.36 ± 0.04 | 0.13 ± 0.02 |
| | 220-230 | | 690 ± 46 | 1.00 ± 0.11 | 1.28 | 11.69 | 1.45 | -6.29 ± 0.14 | 0.28 ± 0.04 | 1.34 ± 0.03 | 0.21 ± 0.03 |
| | 230-240 | 7.39 ± 0.16 | 638 ± 43 | 1.00 ± 0.13 | 1.31 | 11.75 | 1.45 | -7.10 ± 0.94 | 0.41 ± 0.07 | 1.31 ± 0.03 | 0.31 ± 0.05 |
| | 240-250 | | 559 ± 40 | 1.00 ± 0.26 | 1.35 | 11.79 | 1.45 | <-20 ± 0 | 0.53 ± 0.09 | 1.29 ± 0.03 | 0.41 ± 0.07 |
| MBTP9 | 210-220 | | 773 ± 41 | 1.00 ± 0.03 | 1.25 | 11.63 | 1.49 | -6.13 ± 0.09 | 0.26 ± 0.03 | 1.73 ± 0.08 | 0.15 ± 0.02 |
| | 220-230 | | 680 ± 37 | 1.00 ± 0.01 | 1.29 | 11.72 | 1.49 | -6.18 ± 0.10 | 0.36 ± 0.05 | 1.72 ± 0.08 | 0.21 ± 0.03 |
| | 230-240 | 7.07 ± 0.15 | 625 ± 40 | 1.18 ± 0.33 | 1.32 | 11.79 | 1.49 | -6.33 ± 0.17 | 0.44 ± 0.06 | 1.71 ± 0.09 | 0.26 ± 0.04 |
| | 240-250 | | 502 ± 36 | 1.10 ± 0.27 | 1.36 | 11.85 | 1.49 | -6.51 ± 0.24 | 0.56 ± 0.08 | 1.70 ± 0.09 | 0.33 ± 0.05 |